# Metabolic reaction fluxes as amplifiers and buffers of risk alleles for coronary artery disease

Carles Foguet [1,2,3✉], Xilin Jiang[1,2,3,4], Scott C Ritchie[1,2,3,5,6,7], Elodie Persyn[1,2,3], Yu Xu[1,2,3], Chief Ben-Eghan[1,2,3], Henry J Taylor [2,3,8], Emanuele Di Angelantonio[2,3,5,6,9,10], John Danesh[2,3,5,6,9,11], Adam S Butterworth[2,3,5,6,9], Samuel A Lambert [1,2,3,6] & Michael Inouye [1,2,3,5,6,7✉]

## Abstract

**Genome-wide association studies have identified thousands of variants associated with disease risk but the mechanism by which such variants contribute to disease remains largely unknown. Indeed, a major challenge is that variants do not act in isolation but rather in the framework of highly complex biological networks, such as the human metabolic network, which can amplify or buffer the effect of specific risk alleles on disease susceptibility. Here we use genetically predicted reaction fluxes to perform a systematic search for metabolic fluxes acting as buffers or amplifiers of coronary artery disease (CAD) risk alleles. Our analysis identifies 30 risk locus–reaction flux pairs with significant interaction on CAD susceptibility involving 18 individual reaction fluxes and 8 independent risk loci. Notably, many of these reactions are linked to processes with putative roles in the disease such as the metabolism of inflammatory mediators. In summary, this work establishes proof of concept that biochemical reaction fluxes can have non-additive effects with risk alleles and provides novel insights into the interplay between metabolism and genetic variation on disease susceptibility.**

**Keywords** Coronary Artery Disease; Fluxomics; Genome-Scale Metabolic Models; Genomics
**Subject Categories** Cardiovascular System; Genetics, Gene Therapy & Genetic Disease; Metabolism

## Introduction

Genome-wide association studies (GWAS) have identified tens of thousands of single nucleotide polymorphisms (SNPs) associated with disease risk (Sollis et al, 2023). However, variant-to-function (V2F) remains largely unsolved limiting the potential to leverage GWAS results to uncover the underlying disease biology and identify novel therapeutic targets (Nandakumar et al, 2020; Claussnitzer and Susztak, 2021). Genetic variants do not act in isolation but within the context of highly complex biological networks that can modulate the effect of specific alleles on disease susceptibility (Joshi et al, 2021; Lappalainen et al, 2024). Therefore, unveiling non-additive effects between genetic variants and environment or genetic factors is a powerful approach to understanding the functional relationships of genetic variants and their role in disease (Niel et al, 2015; Wu et al, 2023).

Metabolism is a major biological network comprised of metabolites, enzymes, and transmembrane carriers and underlies many processes in health and disease (Frayn, 2010). One of the most direct manifestations of the metabolic phenotype are metabolic fluxes: the rate at which substrates are converted to products in biochemical reactions or transported across compartments in a metabolic network (Zamboni et al, 2015; Niedenführ et al, 2015). Dysregulation of metabolic fluxes can play a major role in disease onset and progression (Oller et al, 2022; Sims and Muyderman, 2010; Tian and Liang, 2021; Lopaschuk et al, 2021). For instance, it is well-established that alterations in lipid metabolic pathways can disrupt the concentration of lipids in blood and promote the formation of atherosclerotic lesions (Libby et al, 2019; Soppert et al, 2020).

Furthermore, metabolic fluxes are particularly attractive as therapeutic targets for perturbation as it has been shown that they can be safely modulated to minimise disease risk and progression. For instance, statins, which are widely prescribed to reduce cardiovascular disease risk, act by reducing the flux of cholesterol synthesis through the inhibition of the enzyme HMG-CoA Reductase (Ward et al, 2019). Statins can be combined with inhibitors of bile acid reabsorption, which can further reduce cholesterol levels by increasing the flux of bile acid synthesis (Mach et al, 2020; Soppert et al, 2020). Similarly, one of the mechanisms of action of the type 2 diabetes drug metformin, which is also reported to have cardioprotective effects (Han et al, 2019), is the reduction of the oxidative phosphorylation flux (Foretz et al, 2023).

[1]Cambridge Baker Systems Genomics Initiative, Department of Public Health and Primary Care, University of Cambridge, Cambridge, UK. [2]British Heart Foundation Cardiovascular Epidemiology Unit, Department of Public Health and Primary Care, University of Cambridge, Cambridge, UK. [3]Victor Phillip Dahdaleh Heart and Lung Research Institute, University of Cambridge, Cambridge, UK. [4]Department of Epidemiology, Harvard T.H. Chan School of Public Health, Boston, MA, USA. [5]British Heart Foundation Centre of Research Excellence, University of Cambridge, Cambridge, UK. [6]Health Data Research UK Cambridge, Wellcome Genome Campus and University of Cambridge, Cambridge, UK. [7]Cambridge Baker Systems Genomics Initiative, Baker Heart and Diabetes Institute, Melbourne, VIC, Australia. [8]Center for Precision Health Research, National Human Genome Research Institute, National Institutes of Health, Bethesda, MD, USA. [9]National Institute for Health and Care Research Blood and Transplant Research Unit in Donor Health and Behaviour, University of Cambridge, Cambridge, UK. [10]Health Data Science Research Centre, Fondazione Human Technopole, Milan, Italy. [11]Department of Human Genetics, the Wellcome Trust Sanger Institute, Wellcome Trust Genome Campus, Hinxton, UK. ✉E-mail: cf545@medschl.cam.ac.uk; mi336@cam.ac.uk

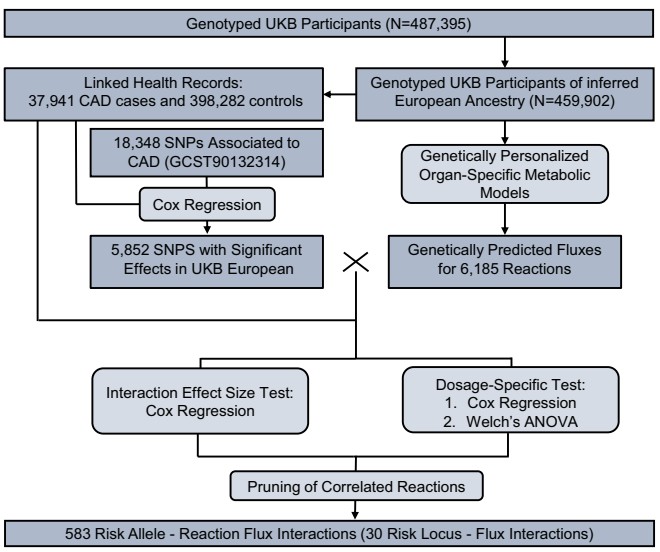

**Figure 1.** Summary of the methods to detect interactions between reaction fluxes and risk alleles on CAD risk.

Notably, intracellular metabolic fluxes cannot be directly measured and instead must be predicted from data integrated into the framework of metabolic networks (Zamboni et al, 2015; Niedenführ et al, 2015). We have previously demonstrated the feasibility and insights that can be gleaned from using a genotyped population-based biobank to estimate metabolic reaction fluxes and perform a fluxome-wide association analysis (FWAS) for coronary artery disease (CAD) (Foguet et al, 2022). This analysis was performed using genetically predicted fluxes derived from the integration of genetically imputed transcript abundances (Gamazon et al, 2015) within the constraints defined by the stoichiometric relationships of enzymes and transmembrane carriers in human genome-scale metabolic networks (Robinson et al, 2020). We found that fluxes through several reactions linked to the metabolism of inflammatory mediators (e.g., histamine and prostaglandins) and polyamines were strongly associated with CAD risk (Foguet et al, 2022).

Since alterations in metabolic function contribute to disease susceptibility and progression, here we set out to test the hypothesis that metabolic fluxes can also act as buffers or amplifiers of the effects of risk alleles. Using UK Biobank (UKB) (Sudlow et al, 2015; Bycroft et al, 2018) as a discovery cohort, we identified numerous interactions between genetically personalized organ-specific fluxes and the dosage of risk alleles on CAD which were then replicated in the All of Us Research Program cohort (All of Us Research Program Investigators et al, 2019; All of Us Research Program Genomics Investigators, 2024). In doing so, we identified numerous instances where reaction fluxes significantly amplify or buffer the effects of genetic variants on CAD risk.

# Results

## Overview of the methods

We estimated genetically personalized metabolic reaction fluxes for 6185 organ-specific reactions in 459,902 UK Biobank participants (Sudlow et al, 2015; Bycroft et al, 2018) of European genetic ancestries

(Manichaikul et al, 2010) (Methods). From hospital episode statistics and cause of death records for these participants, there were 37,941 CAD cases in total (combined prevalent and incident cases) using the PheWAS Catalog definition of coronary atherosclerosis (Wu et al, 2019), and 398,282 non-CAD controls. Using a published genome-wide association meta-analysis performed on over one million participants of European ancestry (Aragam et al, 2022), we extracted a set of 18,348 SNPs associated with CAD risk ($P < 5 \times 10^{-8}$). For downstream analyses, we used the subset of 5852 SNPs which were found significantly associated with CAD in European UKB participants ($P < 5 \times 10^{-8}$; Cox regression model; Methods; Fig. 1).

We next assessed the extent to which each of the 6185 reaction fluxes either buffered (i.e. negative interaction) or amplified (i.e. positive interaction) the effects of the 5852 risk alleles on CAD (Methods). To robustly identify such events, we used the intersection of two complementary approaches with different statistical assumptions. In the first approach, we tested for a significant interaction effect size between risk allele dosage and reaction flux value using a Cox proportional-hazards model for CAD events. Before testing for interaction, the effect of the risk allele was regressed from the reaction flux to control for potential false positives arising from dependencies between genetically predicted flux values and risk allele dosage (Hemani et al, 2021). In the second approach, termed dosage-specific test, we tested the differences in reaction flux effect sizes on CAD risk between individuals carrying different risk allele dosages (Appendix Fig. S1; Methods). In both approaches, we controlled for age, sex, and genetic principal components. We also evaluated the impact of including additional non-genetic cardiovascular risk factors as covariates, namely BMI, systolic blood pressure, and blood levels of LDL-cholesterol, HDL-cholesterol, and triglycerides (Methods) finding that their inclusion did not significantly alter interaction effect estimates (Appendix Fig. S2). The P-values from the interaction model and the dosage-specific test were then adjusted for multiple testing using the Benjamini-Hochberg method (i.e. FDR). A buffering or amplification effect of a reaction flux on a risk allele had to be significant under both approaches (FDR-adjusted P-value < 0.05) to be deemed valid.

## Buffering and amplification of the effect of risk variants by reaction fluxes

In total, we found 583 pairs of SNP-reaction fluxes which were significant (FDR-adjusted P-value < 0.05) in both the interaction effect size test and the dosage-specific test (separately, 669 and 595 SNP–reaction flux pairs were significant, respectively) (Dataset EV1). We observed strong correlations between interaction effect estimates (r = 0.998) and the P-values (r = 0.713) of both approaches (Appendix Fig. S3). Indeed, from the 86 pairs significant for the interaction effect size test but not the dosage-specific test, 82 were significant with an FDR-adjusted P-value < 0.25 in the latter. Similarly, 7 out of the 12 pairs significant with the dosage-specific test but not the interaction effect size test were also borderline significant, with the remaining pairs being instances where the interaction might significantly deviate from linearity.

Of the 583 SNP–reaction flux pairs, 353 displayed buffering (i.e. negative interaction effect size) and 230 displayed amplification (i.e. positive interaction effect size) of the effects of the risk allele by reaction fluxes (Fig. 2). The significant pairs comprised 279 unique SNPs mapped to 8 independent risk loci ($R^2 < 0.6$; Methods) leading to 30 risk locus–reaction flux pairs with significant interaction on disease susceptibility. These interactions

encompassed 18 unique reaction fluxes. Notably, only five of these fluxes had a significant effect on CAD risk in univariate association analysis, indicating that the majority of the metabolic associations here unveiled have their association with CAD masked unless analysed together with common risk alleles (Appendix Fig. S4).

We quantified the linkage disequilibrium (LD) between the 279 SNPs with significant interaction and the expression quantitative trait loci (eQTL) variants for metabolic gene expression used in flux computation. There were some instances where there was a strong LD between risk variants and metabolic eQTLs. This was expected given that metabolic genes are mapped to some of the CAD risk loci. However, if these eQTLs also had strong individual effects on reaction fluxes, they could introduce spurious interactions by correlating genetically predicted fluxes with risk allele dosage. To account for this, we had regressed the effect of risk alleles from reaction fluxes when testing for interactions (Methods). To validate this approach, we assessed the effect of individual eQTLs on the 18 reaction fluxes involved in significant interactions. None of the eQTLs in strong LD with CAD risk variants showed a strong correlation with the reaction fluxes identified as amplifiers or buffers of these variants' effects (Appendix Fig. S5).

## The genomic region encoding Lp(a) and plasminogen is a major site for amplification and buffering

The majority (530 out of 583) of the SNP–reaction flux pairs with significant interaction on CAD risk were mapped to four risk loci in the chromosome 6 genomic region encoding the *LPA* and *PLG* genes (Fig. 3A; Appendix Fig. S6; Dataset EV1). *LPA* codes for apolipoprotein(a) which is the primary constituent of Lp(a) and has been established to have a causal role in the formation of atherosclerotic lesions by promoting lipid accumulation, inflammation, and calcification in the artery wall (Reyes-Soffer et al, 2022; Cho et al, 2013; Van Der Valk et al, 2016). *PLG* encodes plasminogen which can contribute to atherosclerosis by modulating, cell migration, extracellular matrix structure, vascular smooth muscle cell (VSMC) function, and inflammation (Plow and Hoover-Plow, 2004; Rossignol et al, 2006; Kremen et al, 2008).

The flux of prostaglandin E2 transport in both heart and brain tissues strongly amplifies the effect size of a set of risk variants mapped to the risk locus which includes the *LPA* gene (Fig. 3A; Appendix Fig. S6). Prostaglandin E2 is an inflammatory mediator that plays a role in the pathogenesis of atherosclerosis (Gomez et al, 2013). Notably, prostaglandin transport is mediated by *SLC22A1*, *SLC22A2* and *SLC22A3* coded by genes at the *LPA*/*PLG* locus (Nigam, 2018; Robinson et al, 2020). Interestingly, the flux of elongation of arachidonoyl-CoA in adipose tissue has the opposite effect and buffers the effect size of two SNPs in the same risk locus. Arachidonoyl-CoA is the CoA-conjugated form of arachidonic acid which is a precursor for prostaglandin synthesis (Robinson et al, 2020) further reinforcing the role of prostaglandin metabolism in modulating the effect of risk alleles (Fig. 3B). Similarly, variants in the same locus also had their effect size amplified by the flux of histamine synthesis in adipose tissue and a flux of histamine transport into the liver (Appendix Fig. S6). Histamine is also an inflammatory mediator that has been linked to atherosclerosis by modulating inflammation, blood lipids, and lipoprotein fractions (Inouye et al, 2010; Wang et al, 2011).

We also found that fluxes involving polyamine transport in adipose tissue, heart, and skeletal muscle can have buffering or amplification effects in a variant- and tissue-specific manner across the four adjacent risk loci (Fig. 3A; Appendix Fig. S6). Polyamines are a family of pleiotropic compounds that regulate cell proliferation, cell differentiation, and protein synthesis and have a generally protective effect against inflammation and oxidative stress (Xuan et al, 2023). Polyamine-rich diets have been established to protect against cardiovascular disease by countering the age-related myocyte and vascular endothelial dysfunctions (Xuan et al, 2023) while dysregulation of endogenous polyamine metabolism can also lead to pathologies such as cardiac hypertrophy (Meana et al, 2016; Giordano et al, 2010). In adipose tissue, polyamine synthesis is reported to protect against obesity by promoting vascularization and lipolysis (Monelli et al, 2022) while in skeletal muscle it promotes muscle mass growth and regeneration (Lee and Maclean, 2011).

Finally, in the risk locus encoding the *PLG* gene, two intron variants displayed significant interaction on disease risk with the flux through N-Acetylglucosamine 2-Epimerase in the liver (Fig. 3A). This reaction is catalysed by a homodimer of the renin-binding protein (RNBP) (Robinson et al, 2020; Takahashi et al, 2006). The formation and stabilization of its catalytically active form prevents the formation of a heterocomplex with renin, which sequesters the latter and inhibits the renin-angiotensin system (Takahashi et al, 2006) (Fig. 3C). The renin-angiotensin system increases blood pressure and has been linked to atherosclerosis (Wu et al, 2018).

## Transport of stearidonoyl-carnitine amplifies the effect size of variants mapped to the *BCAR1/CFDP1* risk locus

In the *BCAR1/CFDP1* risk locus, there was evidence of risk allele amplification by the flux of mitochondrial transport of stearidonoyl-carnitine in skeletal muscle, with 38 risk variants showing significant interaction with this flux (Fig. 4A; Dataset EV1). Polymorphisms within this region have been associated with carotid intima-media thickness (i.e. a marker of subclinical atherosclerosis) and CAD risk with *BCAR1* identified as the likely causal gene (Gertow et al, 2012; Boardman-Pretty et al, 2015). *BCAR1* regulates cell migration, proliferation and apoptosis and is essential for cardiovascular development in embryogenesis (Barrett et al, 2013). In particular, *BCAR1* is a major regulator of VSMC function and it has been theorized that *BCAR1*'s contribution to the formation of atherosclerotic lesions arises from its role in VSMC migration (Camacho Leal et al, 2015).

Stearidonoyl-carnitine is the carnitine-conjugated form of stearidonic acid (SDA), an omega-3 polyunsaturated fatty acid that can be efficiently metabolized to eicosapentaenoic acid (EPA; Fig. 4B) (Whelan et al, 2012). EPA has well-established anti-inflammatory and cardioprotective effects (Hirafuji et al, 2003; Khan et al, 2021) and has been shown to inhibit the progression of atherosclerotic lesions (Mita et al, 2007). Likewise, SDA has also been found to have anti-inflammatory effects in cell and animal models (Whelan et al, 2012; Sung et al, 2017). However, SDA levels in the blood have also been associated with an increased risk of cardiovascular pathologies such as hypertension (Al Ashmar et al, 2024) or thrombosis (Li et al, 2023) suggesting a context-specific role in disease.

## Galactose transport into the brain amplifies the effect size of variants mapped to the *SMARCA4* risk locus

A set of 19 variants mapped to a risk locus encoding the *SMARCA4* gene, were identified as having their effect size amplified by the flux of the sodium-coupled galactose transport in the brain (Fig. EV1; Dataset EV1). *SMARCA4* codes for a chromatin-remodelling factor

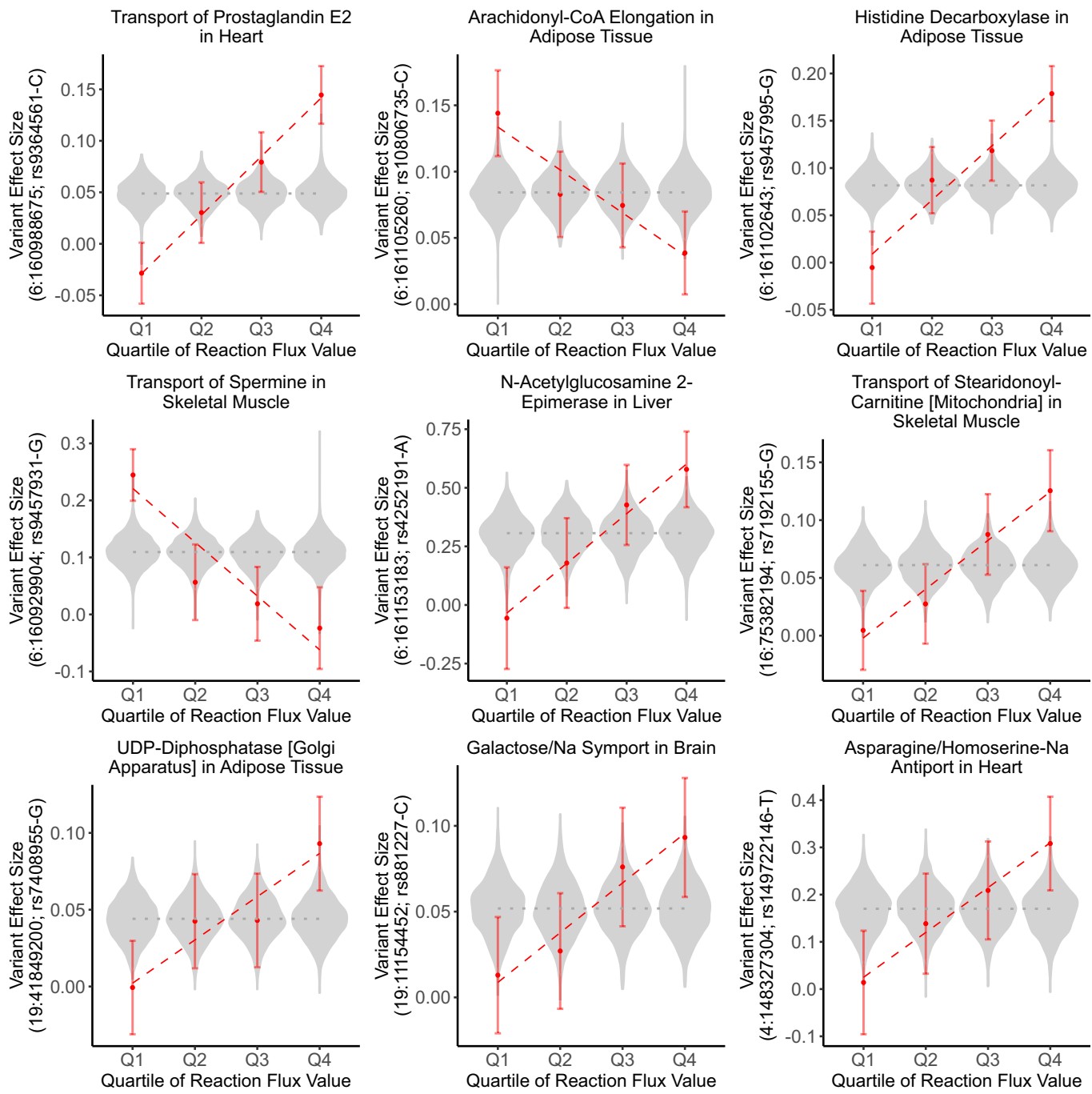

**Figure 2. Representative examples of buffering and amplification of variant effect sizes by reaction fluxes on CAD risk.**

For each plotted SNP-reaction pair, the UKB participants of European genetic ancestries were quartile-binned according to reaction flux value and variant effect sizes on CAD risk were estimated within each flux quartile. The dashed red lines indicate the linear regression of variant effect sizes per flux quartile. Violin plots indicate the distribution of variant effect size on CAD risk for all other reaction fluxes. The dotted grey line indicates variant effect size in all the analysed UKB participants. Variant effect sizes are expressed as Log(Hazard Ratio) per risk allele dosage and were computed with Cox regression. Error bars indicate the 95% confidence intervals for variant effect size estimates. Genome coordinates correspond to the GRCh37 genome assembly.

that has a major role in transcriptional regulation, DNA repair, and cell proliferation in a wide range of processes and tissues (Trotter and Archer, 2008). The *SMARCA4* gene is adjacent to *LDLR*, a well-established risk locus for CAD, but it has LDLR-independent roles in cardiovascular disease risk (Cave et al, 2023). For instance, *SMARCA4*

has been reported to mediate vascular calcification (Wang et al, 2019), inflammation (Cave et al, 2023) and myocardial proliferation (Xiao et al, 2016). Indeed, the variants showing significant interaction with galactose transport are in a different LD block than those linked to *LDLR* (Fig. EV1).

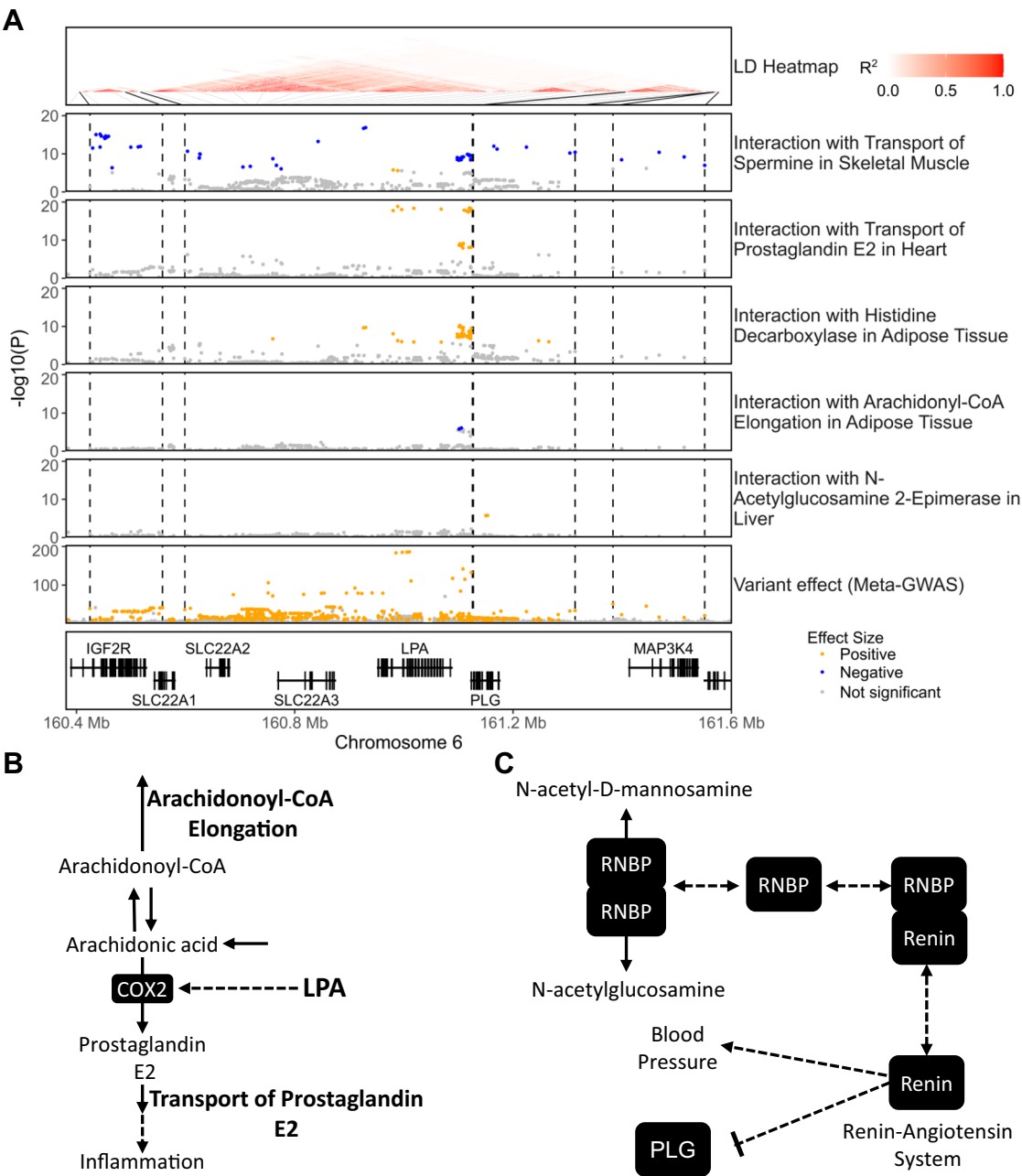

**Figure 3. Examples of amplification and buffering at the *LPA/PLG* loci.**

(A) Regional association plots showing the −log10(P-value) for interaction and variant effect sizes on CAD risk. The complete set of interactions for this region is shown in Appendix Fig. S6. P-values for interaction effect sizes were computed with Cox regression (Methods) and variant effect sizes were obtained from a published GWAS meta-analysis (Aragam et al, 2022). The LD heatmap indicates the pairwise LD for SNPs with genome-wide significant effect size on CAD in analysed UKB participants. Dashed black lines indicate the limits of LD blocks (R² > 0.6) used to define independent risk loci. To facilitate visualization, only LD blocks with variants involved in significant interactions are highlighted. Genome coordinates correspond to the GRCh37 genome assembly. (B) Potential mechanism of interaction between reactions of prostaglandin metabolism and *LPA*. LPA induces the expression of the enzyme COX2 which catalyses the formation of prostaglandin E2 from arachidonic acid. Conversely, the elongation of arachidonoyl-CoA diverts arachidonic acid away from prostaglandin synthesis. (C) Potential mechanism of interaction between N-Acetylglucosamine 2-Epimerase and plasminogen. This reaction is catalysed by a homodimer of RNBP, which can also form a heterodimer with renin inhibiting the renin-angiotensin system. The renin-angiotensin system can suppress plasminogen activation. Solid arrows denote metabolic reactions or transport processes and dashed lines other functional relationships.

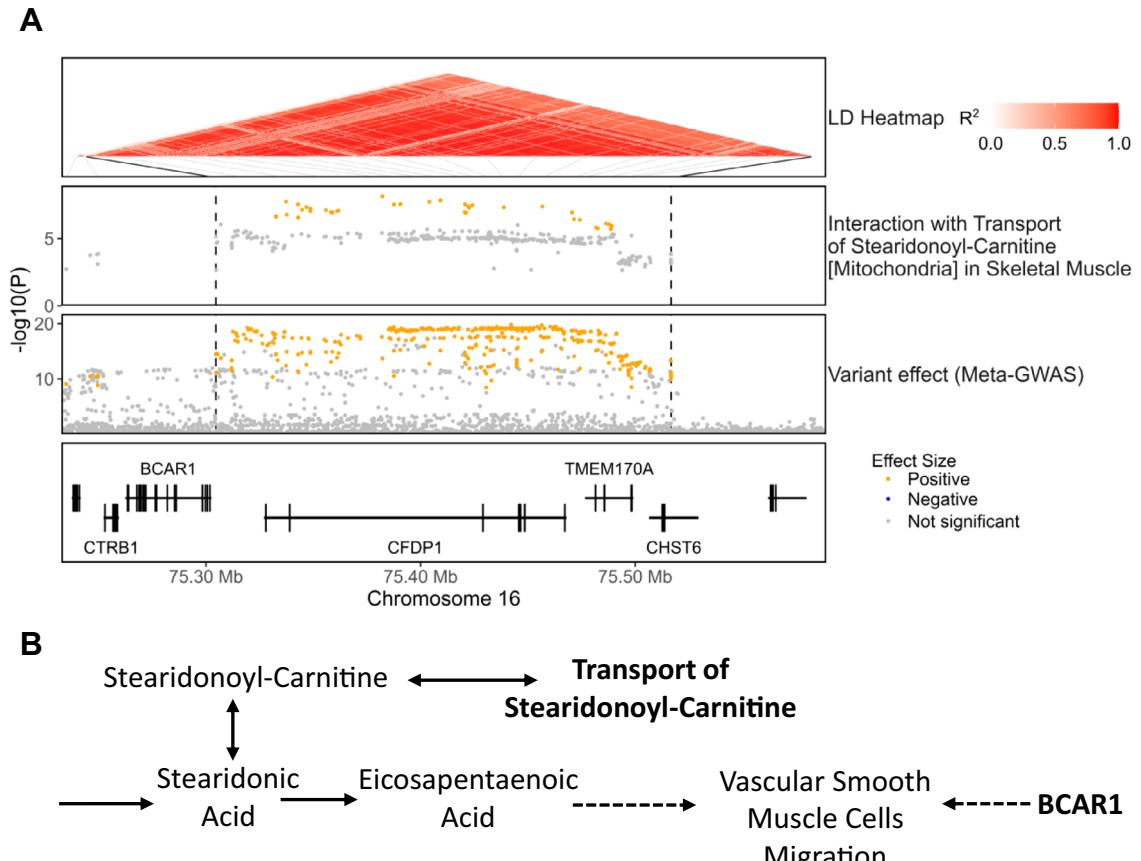

**Figure 4. Transport of stearidonoyl-carnitine amplifies the effect size of risk variants of the *BCAR1/CFDP1* locus.**

(A) Regional association plots showing the −log10(*P*-value) for interaction and variant effect sizes on CAD risk. (B) Potential mechanism of interaction between the transport of stearidonyl-carnitine and risk variants linked to *BCAR1*. Solid arrows denote metabolic reactions or transport processes and dashed lines other functional relationships, namely activation of the migration of vascular smooth muscle cells.

## UDP-Diphosphatase amplifies the effect size of variants mapped to the TGF-β-risk locus

In the TGF-β risk locus, there was evidence of risk allele amplification by the flux through UDP-diphosphatase (Golgi apparatus) in adipose tissue, with two intron variants of *TGFB1* showing significant interaction on disease risk with this flux (Fig. 5A; Dataset EV1). *TGFB1* codes for a ligand of the Transforming Growth Factor β (TGF-β) family which regulates a wide range of biological processes such as morphogenesis, tissue homeostasis, and inflammation in a context-specific manner (Massagué, 2012). TGF-β plays a major role in the cardiovascular system by regulating the proliferation, differentiation and function of endothelial, smooth muscle, and immune cells, and its dysregulation can lead to a wide range of cardiovascular diseases such as atherosclerosis (Goumans and Ten Dijke, 2018). TGF-β signalling is also linked to adiposity through the regulation of adipocyte differentiation and oxidative metabolism (Zamani and Brown, 2011), and *TGFB1* is overexpressed in obesity (Samad et al, 1997; Fain et al, 2005).

UDP-diphosphatase plays a major role in the Golgi apparatus, which is the primary location of protein and lipid glycosylation. Nucleotide sugars (such as UDP-sugars) are transported from the cytosol to the Golgi's lumen where they act as donors for a glycosylation process releasing the nucleotide diphosphate (e.g. UDP) (Stanley, 2011). Notably,

the transport of UDP-sugars to Golgi involves an antiport with luminal UMP (Caffaro and Hirschberg, 2006; Sesma et al, 2009). As such, UDP-Diphosphatase contributes to glycosylation by catalysing the hydrolysis of UDP to UMP hence enabling the UMP-dependent transport of nucleotide sugars (Stanley, 2011; Caffaro and Hirschberg, 2006). In addition, the UMP-dependent transport of nucleotides-sugars to the Golgi's lumen is also reported to contribute to the vesicle-based release of UDP-sugars to the extracellular space (Sesma et al, 2009) (Fig. 5B).

## Amino acid transport amplifies the effect of a risk variant upstream of the *EDNRA* gene

A significant positive interaction on CAD risk was detected between a risk variant, mapped upstream of the *EDNRA* gene, and the flux of an amino acid transport process in the heart (i.e. the sodium-coupled exchange of homoserine and asparagine) (Fig. EV2; Dataset EV1). *EDNRA*, which is expressed in VSMCs and cardiomyocytes, codes for the receptor for endothelin-1. Endothelin-1, acting through the activation of EDNRA, is a potent vasoconstrictor with a well-established role in cardiovascular diseases (Karmazyn, 2017). *EDNRA* expression in vascular tissue and endothelin-1 levels in the blood are altered in atherosclerosis (Fan et al, 2000; Stölting et al, 2020) and EDNRA antagonists can attenuate the progression of coronary atherosclerosis lesions (Yoon et al, 2013). The

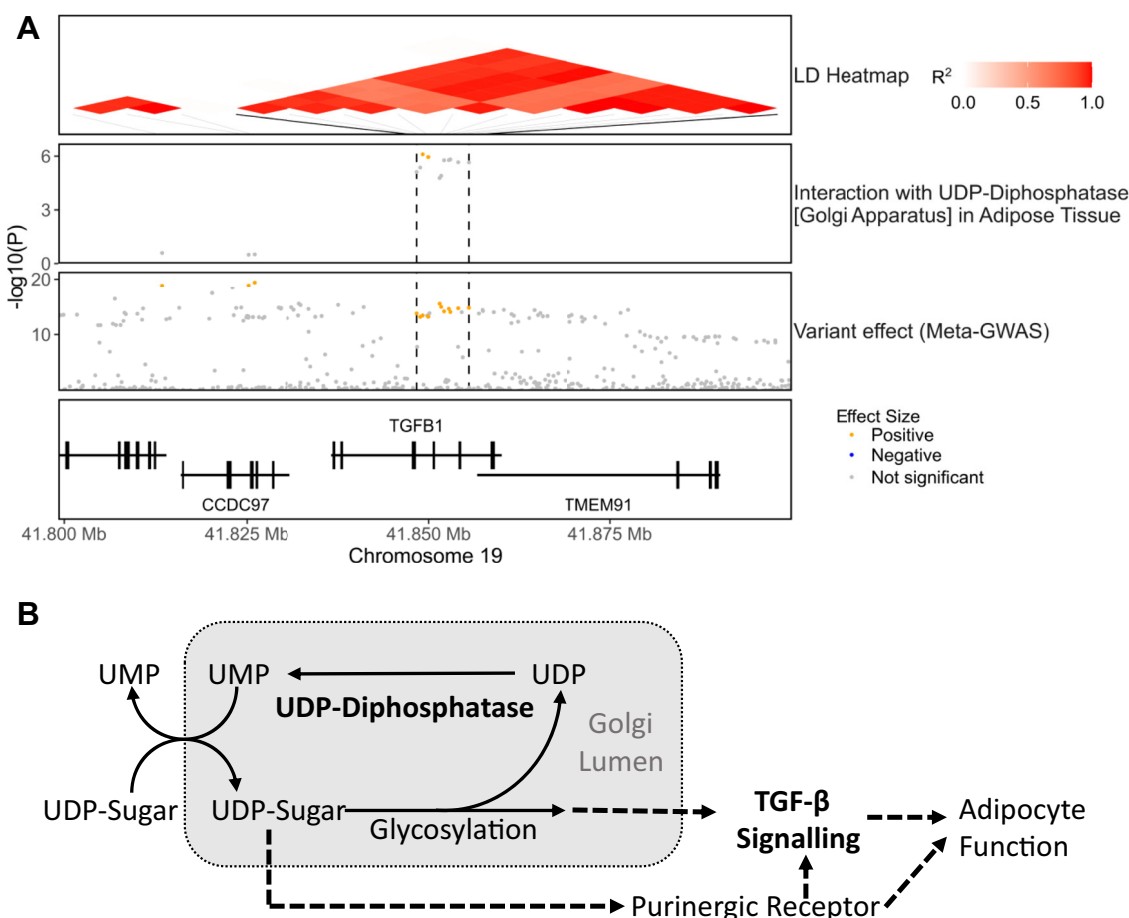

**Figure 5. UDP-diphosphatase amplifies the effect size of CAD risk variants within the *TGF-β* risk locus.**

(A) Regional association plots showing the −log10(*P*-value) for interaction and variant effect sizes on CAD risk. (B) Potential mechanism of interaction between UDP-diphosphatase and risk variants linked to *TGFB1*. Solid arrows denote metabolic reactions or transport processes and dashed lines other functional relationships.

concentrations in blood of both asparagine and homoserine have been associated with cardiovascular disease risk (Hasokawa et al, 2012; Ottosson et al, 2018; Patel et al, 2020; Aa et al, 2021), however, the mechanism of their effect remains largely unknown.

## Buffering and amplification of the risk of myocardial infarction

Myocardial infarction (MI) is caused by a sudden blockage of blood flow to the myocardium, primarily due to coronary atherosclerosis with or without a blood clot (Ojha and Dhamoon, 2024). Given the close relationship between MI and CAD and that the metabolic reactions we assess here may not a priori be involved in hard outcomes like MI, we investigated the extent to which the identified instances of buffering and amplification of risk allele penetrance on CAD risk could also be relevant for MI risk.

On the UKB participants of European genetic ancestries, we identified 36,007 MI cases and 423,629 controls using an established definition of MI (Inouye et al, 2018). As expected, there was a substantial overlap of cases and controls between MI and coronary atherosclerosis (Appendix Table S1). Risk variant and reaction flux effect sizes, estimated using Cox regression, were highly correlated

between coronary atherosclerosis and MI (Appendix Fig. S7A–D). However, reaction effect sizes for MI risk were on average lower than the equivalent effect sizes derived for coronary atherosclerosis.

We evaluated buffering and amplification effects between risk variants and risk allele dosage on MI risk. We tested the same SNP–reaction flux pairs that had been evaluated with coronary atherosclerosis to facilitate the comparison of the results. For MI, there were 426 pairs significant with both interaction effect size and dosage-specific tests (FDR-adjusted *P*-value < 0.05) (Dataset EV2). Taking into account the LD structure of risk SNPS, these represented 26 risk locus–reaction flux pairs with evidence of significant amplification or buffering of risk alleles by reaction fluxes. Out of the 583 significant SNP-flux pairs for coronary atherosclerosis, 360 were also significant in MI (20 risk locus–reaction flux pairs) with 91 additional SNP-flux pairs (4 additional risk locus–reaction flux pairs) being borderline significant (FDR-adjusted *P*-value < 0.25 for both the interaction effect size test and the dosage-specific tests). Overall, interaction effect sizes were also strongly correlated between MI and coronary atherosclerosis (Appendix Fig. S7E–H). Like in CAD, the interaction estimates were also robust to the inclusion of additional cardiometabolic covariates (Appendix Fig. S2).

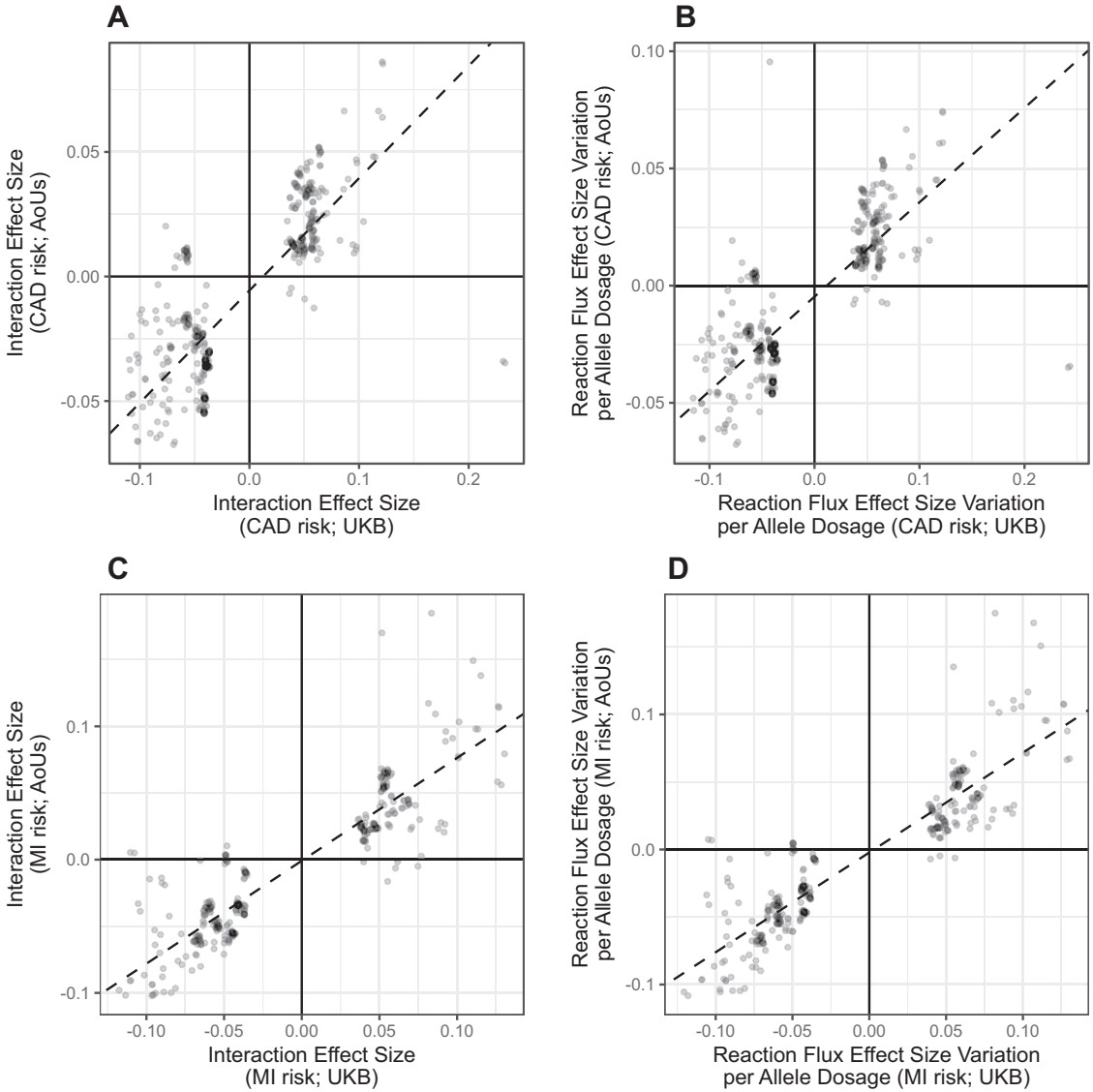

**Figure 6.  Validation of Interaction effects in the All of Us (AoUs) cohort.**

The pairs of SNP-reaction flux with significant interaction on coronary atherosclerosis (CAD) or myocardial infarction (MI) risk identified in UKB were evaluated in the AoUs cohort using two complementary methods. In the first method, interaction is measured as the coefficient of the interaction term between the risk allele and reaction flux on CAD (**A**) or MI risk (**C**). The second method quantifies interactions as the variation of reaction flux effect sizes on CAD (**B**) or MI (**D**) risk across risk allele dosages (Methods). The dashed line indicates the linear regression of interaction effects between UKB and AoUs. Interaction effects are expressed as the log(Hazard Ratio) per standard deviation of log(flux value) and allele dosage on CAD or MI risk.

Notably, there were no significant interactions on MI risk between the transport of stearidonoyl-carnitine and the risk locus of *BCAR1/CFDP1* nor UDP-diphosphatase and the *TGFB1* risk locus. Effect sizes for variants mapped to these loci estimated with Cox regression were also slightly lower in MI compared to coronary atherosclerosis (Fig. EV3). Conversely, two loci had significant buffering of disease risk alleles by reaction fluxes in MI and not in CAD: effect sizes of 15 risk alleles mapped to the *PDE5A/MAD2L1* locus, and 13 risk alleles mapped to the *MAP1S/FCHO1* locus were buffered by the flux of orotate phosphoribosyltransferase in heart and uracil transport in skeletal muscle, respectively (Fig. EV4). Both reactions are functionally part of pyrimidine metabolism, which has been implicated in MI (Li et al, 2022; Mazzola et al, 2008;

Yitzhaki et al, 2005). Notably, PDE5A is also linked to nucleotide metabolism and its inhibition is well-established to have cardio-protective effects in MI (Hutchings et al, 2018; Li et al, 2021).

## External validation in the All of Us cohort

To assess the reproducibility of the buffering or amplification effects identified in the UKB cohort, we conducted an external validation using data from the All of Us (AoUs) cohort (All of Us Research Program Investigators et al, 2019; All of Us Research Program Genomics Investigators, 2024). Genetically personalized fluxes were computed for all AoUs participants of European genetic ancestries ($N = 118,058$). From the linked electronic health records, we identified 14,117 CAD cases and

75,204 non-CAD controls using the PheWAS Catalog definition of coronary atherosclerosis (Wu et al, 2019). Then, the 583 SNP-reaction fluxes pairs with amplification or buffering effects in CAD risk discovered in UKB were evaluated for interaction effects in AoUs using both the interaction effect size test and the dosage-specific test (Methods). Despite a smaller sample size and fewer CAD cases in AoUs compared to UKB, interaction effect estimates were highly consistent between both cohorts (r = 0.796 for the interaction effect size test and r = 0.80 for the dosage-specific test; Fig. 6A,B). Out of the 583 pairs discovered in UKB, 253 were significant (FDR-adjusted P-value < 0.05) in AoUs with either the interaction effect size or dosage-specific test with an additional 131 pairs being borderline significant (FDR-adjusted P-value < 0.25; Dataset EV1). Furthermore, 548 pairs of the 583 tested pairs (94%) showed consistent interaction effect size directions between UKB and AoUs.

Following the same approach, the 426 instances of amplification and buffering of MI risk discovered in UKB were also evaluated in AoUs (7218 MI cases and 86,055 non-MI controls). Consistent with a lower phenotype heterogeneity between cohorts, the resulting interaction estimates for MI showed even greater consistency than those for CAD (r = 0.9 for the interaction effect sizes test and r = 0.91 for the dosage-specific test; Fig. 6C,D). A total of 168 SNP-reaction fluxes pairs showed significant interactions in AoUs with either the interaction effect size or dosage-specific test and 175 additional pairs were borderline significant (Dataset EV2). Furthermore, a total of 410 (96%) SNP-reaction fluxes pairs had consistent directionality of interaction effects estimates with UKB.

# Discussion

Here, we used genetically personalized organ-specific metabolic fluxes to study the interaction between risk variants and biochemical reactions on disease risk using CAD as a case study. Our analysis identified 18 metabolic reaction fluxes that can amplify or buffer risk allele effect sizes at 8 well-established CAD risk loci unveiling a total of 30 risk locus–reaction flux pairs with significant interaction on disease susceptibility. Most of such reactions involved metabolic processes with known roles in atherosclerosis such as inflammation or fatty acid metabolism. We also find that the majority of the interactions detected for CAD were also relevant for MI. Furthermore, we establish that most of the interaction signals discovered in UKB can be replicated in the external cohort AoUs.

We identify that the genomic region of LPA/PLG, a well-known locus for CAD risk (Schunkert et al, 2011; Aragam et al, 2022), is a major site of interaction between risk variants and reaction fluxes. For instance, we find that a set of variants in the LPA and PLG risk loci have their effect size amplified by the flux of reactions involved in the synthesis or transport of histamine and prostaglandin E2, two inflammatory mediators that can contribute to the formation of atherosclerotic lesions (Wang et al, 2011; Gomez et al, 2013). In our previous work (Foguet et al, 2022), we identified that some of those reaction fluxes were associated with CAD risk, here we show that this effect can be further amplified or buffered by the dosage of specific risk alleles within the LPA/PLG region. Inflammation is one of the mediators of pathogenicity of LPA and PLG (Plow and Hoover-Plow, 2004; Cho et al, 2013; Van Der Valk et al, 2016) and the former has been reported to induce the expression of cyclooxygenase-2, which catalyses the first step of prostaglandin E2 synthesis (Cho et al, 2013). Hence, the identified interactions may reflect a mechanism where LPA or PLG variants that increase inflammation have their effect on CAD risk amplified by a high capacity to transport prostaglandin E2 or histamine

across cellular membranes. Conversely, the flux of elongation of arachidonoyl-CoA in adipose tissue would attenuate this effect by diverting arachidonic acid away from prostaglandin synthesis (Fig. 3B).

Notably, most interactions were also instances where the effect of a biochemical reaction flux on CAD risk only becomes apparent when analysed in conjunction with risk variants. For instance, neither the flux through N-Acetylglucosamine 2-Epimerase in the liver, the transport of stearidonoyl-carnitine in skeletal muscle nor UDP-diphosphatase in adipose tissue have a significant effect on disease risk when analysed in univariate analysis. However, the interaction analysis reveals that such fluxes can significantly amplify the effect size of risk variants mapped to the PLG, BCAR1 and TGFB1 risk loci, respectively, providing insights into their roles in cardiovascular disease. For instance, the reaction N-Acetylglucosamine 2-Epimerase is mediated by an enzyme that moonlights as an inhibitor of renin (Takahashi et al, 2006) and the formation of its catalytically active form prevents it from inhibiting the renin-angiotensin system (Wu et al, 2018), a known regulator of plasminogen (Vaughan et al, 1995; Brown et al, 1998), suggesting a potential mechanism of interaction with variants of the plasmino-gen risk locus (Fig. 3C). Concerning the transport of stearidonoyl-carnitine, the transport of acylcarnitines to mitochondria is the limiting step for mitochondrial β-oxidation (Lehner and Quiroga, 2016), and hence this reaction might be affecting CAD risk by modulating the bioavailability of SDA and other polyunsaturated fatty acids. In this regard, one of the cardioprotective actions of omega-3 polyunsaturated fatty acids is the inhibition of VSMC proliferation and migration (Hirafuji et al, 2003; Yin et al, 2020), thus providing a link to BCAR1 which is also reported to modulate VSMC function (Camacho Leal et al, 2015) (Fig. 4B). Similarly, UDP-diphosphatase is reported to play a major role in the transport of nucleotide-sugars to the Golgi apparatus (Stanley, 2011; Caffaro and Hirschberg, 2006) and hence can potentially modulate both protein glycosylation and the vesicle-based release of UDP-sugars (Sesma et al, 2009). The formation of the functional form of TGF-β, and many of the other proteins involved in TGF-β signalling, requires glycosylation in the Golgi apparatus (Nüchel et al, 2018; Zhang et al, 2021) suggesting a potential avenue of interaction. In addition, the release of UDP-sugars, acting through the activation of purinergic receptors, can modulate adipocyte differentiation, lipolysis, and inflammation within the adipose tissue (Jain and Jacobson, 2022). Indeed, there is ample evidence of crosstalk between TGF-β signalling and purinergic receptors (Chen et al, 2022; Mederacke et al, 2022; Borges da Silva et al, 2020) suggesting a second potential mechanism of interaction (Fig. 5B).

Our study has limitations. First, the necessary use of fluxes simulated from genetically imputed transcript abundances (Foguet et al, 2022) cannot account for the influence of environmental factors on reaction activities. However, this is also advantageous as it minimises confounding by environment-driven factors (e.g. diet, lifestyle, or BMI) on reaction fluxes and disease risk associations. Notably, we found that the detected interactions were not attenuated if direct cardiometabolic measures were included in the analysis. Furthermore, using genetically predicted fluxes enabled us to tap into the large sample sizes arising from the hundreds of thousands of genotyped individuals in large prospective cohorts such as the UKB and AoUs. Second, the genetically imputed fluxes used in this analysis do not explicitly account for the effect of coding variants and splicing variants. There is a lack of systematic quantitative annotation on their

impact on enzyme activities which presently limits our capacity to incorporate their effects on our metabolic flux modelling framework (Foguet et al, 2022) but hopefully will enhance it in the future. Third, the experimental validation of reaction flux amplification/buffering on human alleles was not feasible. Such validation would face significant challenges and would need to be tackled on a case-by-case basis as the variant-to-function (V2F) gap makes it difficult to establish the precise molecular mechanisms of action for non-coding variants (Nandakumar et al, 2020; Claussnitzer and Susztak, 2021). Nevertheless, based on existing literature and established biological pathways, we propose mechanisms that could serve as the starting point for future research.

In conclusion, this work establishes proof of concept that biochemical reaction fluxes can modulate the effect of disease risk alleles and highlights the importance of considering the burden of risk variants to understand the contribution of metabolism to cardiovascular disease susceptibility. Given that disease-associated metabolic processes represent potential targets against disease, such findings have important implications for personalized medicine as they highlight that the therapeutic efficacy of targeting specific metabolic pathways may depend on each individual's genetic background.

# Methods

### Reagents and tools table

| Reagent/Resource | Reference or Source | Identifier or Catalog Number |
|---|---|---|
| N/A | | |
| **Experimental models** | | |
| N/A | | |
| **Recombinant DNA** | | |
| N/A | | |
| **Antibodies** | | |
| N/A | | |
| **Oligonucleotides and other sequence-based reagents** | | |
| N/A | | |
| **Chemicals, Enzymes and other reagents** | | |
| **Software** | | |
| Python 3.6.8 | https://www.python.org/ | |
| Lifelines v0.26.3 | https://lifelines.readthedocs.io (Davidson-Pilon, 2019) | |
| PheTK-0.1.37 | https://github.com/nhgritctran/PheTK (Tran et al, 2024) | |
| Cobrafunctions v1 | cfoguet/cobrafunctions (Foguet et al, 2022) | |
| R 4.3.1 | https://cran.r-project.org/ | |
| karyoploteR | https://bioconductor.org/packages/release/bioc/html/karyoploteR.html (Gel and Serra, 2017) | |
| Plink 1.9 | https://www.cog-genomics.org/plink/ (Chang et al, 2015) | |
| **Other** | | |
| Organ-Specific Genome-Scale Metabolic Models | https://github.com/cfoguet/cobrafunctions/tree/main/inputs/organ_specific_models (Foguet et al, 2022) | |
| Gene Expression Elastic Net Models (GTEx v8) | https://predictdb.org/ (Gamazon et al, 2015) | |
| UK Biobank | https://www.ukbiobank.ac.uk/ (Sudlow et al, 2015; Bycroft et al, 2018) | |
| All of Us Research Program | https://allofus.nih.gov/ (All of Us Research Program Investigators et al, 2019; All of Us Research Program Genomics Investigators, 2024) | |

## UK Biobank

UKB is a cohort of approximately 500,000 participants from the general UK population (https://www.ukbiobank.ac.uk/). Participants were between age 40 and 69 at recruitment (median 58 years of age; 54% women) and accepted an invitation to attend one of the assessment centres that were established across the United Kingdom between 2006 and 2010 (Sudlow et al, 2015). We used the version 3 release of the UK Biobank genotype data (Bycroft et al, 2018) (https://biobank.ndph.ox.ac.uk/showcase/label.cgi?id=263), which had been imputed to the UK10K/1000 genomes and haplotype reference consortium (HRC) (Loh et al, 2016) panels.

Genotyped UKB participants were assigned a genetic ancestry using KING (Manichaikul et al, 2010). Briefly, genotyped UKB samples were projected to the 10 main genetic principal components computed from samples from the 1000 genome project with known superpopulation groups (American, East Asian, European, and South Asian). Using this projection, KING uses a support-vector-machine-based method to infer the most likely ancestral group of each sample (Manichaikul et al, 2010). UKB participants were assigned to European Ancestry if the probability of belonging to that group was estimated to be 95% or higher.

## All of Us Research Program

The All of Us Research Program (AoUs) is a longitudinal cohort that aims to collect comprehensive health data from one million individuals in the USA. Participants are currently required to be ≥18 years of age and reside in the USA or a US territory (All of Us Research Program Investigators et al, 2019). We used the whole genome sequencing (WGS) data available in the curated data repository version 7. AoU participants of European genetic ancestries were identified from the ancestry assignments provided by AoUs. Similarly to the approach we used in UKB, AoUs had used a reference panel and genetic principal component to assign genetic ancestries to its participants. The protocols for QC of the WGS data and genetic ancestry inference are available in the AoUs portal (https://support.researchallofus.org/hc/en-us/articles/27633757470228-All-of-Us-Genomic-Quality-Report) and summarized in the article describing the v7 genomic data release (All of Us Research Program Genomics Investigators, 2024).

## Genetically personalized organ-specific fluxes

Genetically personalized organ-specific fluxes were computed as previously described (Foguet et al, 2022). Briefly, organ-specific metabolic models for adipose tissue, brain, heart, liver and skeletal muscle were extracted from the Harvey/Harvetta whole-body models (Thiele et al, 2020) and ported to HUMAN1 (Robinson et al, 2020), the latest reconstruction of human metabolism. The GIM3E (Schmidt et al, 2013) algorithm was then used to compute an average reaction flux distribution for each organ consistent with average organ-specific transcript abundances obtained from GTEx (GTEx Consortium, 2013). In parallel, genotype data was used to impute personalized organ-specific transcript abundances for UKB and AoUs participants using the elastic net models from PredictDB (Gamazon et al, 2015). Finally, the quadratic metabolic transformation algorithm (qMTA) was used to integrate the individual-

level organ-specific transcript abundances and the average reaction flux distribution and compute genetically personalised organ-specific reaction flux values for each analysed organ. Flux values for each reaction were log2-transformed and standardized to zero-mean and unit variance.

## Definition of coronary artery disease

Cases of CAD were identified using the definition of coronary atherosclerosis (Phecode 411.4) of the PheWAS Catalog (version 1.2) (Wu et al, 2019). Namely, cases were defined by the presence of any of the constitutive ICD9 (411.81, 414.0, 414.01, 414.02, 414.03, 414.04, 414.05, 414.2, 414.3, 414.4, 996.03 or V45.81, V45.82) and ICD10 (I24.0, I25.1, Z95.1 or Z95.5) codes in hospital episode statistics or death records. In addition, non-cases with any of the constitutive ICD codes of ischemic heart disease (Phecode 410-414.99) were excluded from the controls (Wu et al, 2019). In UKB participants of European genetic ancestries, we identified 37,941 CAD cases and 398,282 controls. In AoUs participants of European genetic ancestries, there were 14,117 CAD cases and 75,204 controls.

In both cohorts, the earliest coded or reported date for disease was converted to the age of phenotype onset. Controls were censored according to the maximum follow-up of the health linkage data (UKB: October 31, 2022; AoUs: July 1, 2022) or the date of death (Tran et al, 2024).

## Definition of myocardial infarction

MI was defined as evidence of a fatal or nonfatal myocardial infarction or major coronary surgery (Inouye et al, 2018). In both UKB and AoUs, myocardial infarction was defined by the presence of ICD-9 codes 410–412, ICD-10 codes I21–I24 or I25.2 in hospital episode statistics or cause of death records or reporting a heart attack during the verbal interview at UKB enrolment (Self-report field 6150 and Self-report field 20002). Major coronary surgery was defined by ICD-9 code V45.81 and ICD-10 code Z95.1. In UKB, major coronary surgery was also identified by OPCS-3 codes 309.4 or 884, OPSC-4 codes K40–K46 or reporting a coronary angioplasty, coronary artery bypass grafts or triple heart bypass at the verbal interview (Self-report field 20004) (Inouye et al, 2018). Similarly, surgery in AoUs was identified using concept sets related to coronary surgery in the domain "procedures". In UKB participants of European ancestries, we identified 36,007 cases and 423,629 controls. In AoUs participants of European genetic ancestries, there were 7218 MI cases and 86,055 controls.

In both cohorts, the earliest coded or reported date for disease or surgery was converted to the age of phenotype onset. Controls were censored according to the maximum follow-up of the health linkage data (UKB: October 31, 2022, AoUs: July 1, 2022) or the date of death(Tran et al, 2024).

## Risk variant selection

A set of variants associated with CAD risk was obtained from a published genome-wide association meta-analysis performed on over one million participants of European ancestry(Aragam et al, 2022). Summary statistics from this study were downloaded from the GWAS catalog (GCST90132314) (Sollis et al, 2023) and they

identified 18,348 biallelic SNPs with genome-wide significance ($P < 5 \times 10^{-8}$).

We anticipated that only a subset of these variants might have significant effects in our cohort. As such, we tested the effect of these variants on CAD risk in the European subset of UKB. For this, we used a similar model and assumptions that we would subsequently use for the amplification and buffering analysis (i.e., Cox proportional-hazards model using age as the time scale stratifying by sex and genotyping array and using the first 10 genetic principal components as covariates). We selected the 5852 variants that had genome-wide significance in our cohort as candidate variants to test for amplification and buffering of disease risk by reaction fluxes.

## Identifying interactions between reaction fluxes and risk variants on disease risk

To robustly identify instances of buffering/amplification between reaction fluxes and risk variants, we used two complementary methods to detect interaction: an interaction effect size test and a dosage-specific test.

In the interaction effect size test, for each pair of reaction flux values and risk allele dosages, we test for a significant interaction term using a Cox proportional-hazards model with age as the time scale for disease risk:

$$h(t) = h0(t) \cdot \exp(\beta_{SNP} \cdot SNP + \beta_{Flux} \cdot Flux + \beta_{SNP^2} \cdot SNP^2 + \beta_{Flux^2} \cdot Flux^2 + \beta_{SNP:Flux} \cdot SNP \cdot Flux)$$

where,

$h(t)$ is the hazard function defining the risk of disease at age $t$.
$h0$ is the baseline hazard.
$SNP$ is the dosage of the risk allele.
$Flux$ is the adjusted reaction flux value. Before testing for interaction, flux values are adjusted with linear regression to remove any potential effects of $SNP$ over the reaction flux value.
$\beta_{SNP}, \beta_{Flux}, \beta_{SNP^2}, \beta_{Flux^2}$ are the first and second-order effect sizes for SNP dosage and flux value.
$\beta_{SNP:Flux} SNP \cdot Flux$ is the interaction effect size between SNP dosage and flux value.

The model was fitted using the CoxPHFitter function from the lifelines python package (Davidson-Pilon, 2019). The significant interaction effect on disease risk was evaluated with a two-tailed Wald test for the interaction effect size.

In the dosage-specific effect size test, for each pair of reaction flux values and risk alleles, we split the analysed UKB participants based on risk allele dosage (0, 1 or 2) and estimated the effect of reaction flux value on CAD risk within each allele dosage using a Cox proportional hazards model with age as the time scale.

$$h_{SNP_i}(t) = h0_{SNP_i}(t) \cdot \exp(\beta_{Flux,SNP_i} \cdot Flux_{SNP_i})$$

where,

$h_{SNP_i}(t)$ is the hazard function defining the risk of disease at age $t$ in individuals with dosage $i$ of the risk allele.
$h0_{SNP_i}$ is the baseline hazard in individuals with dosage $i$ of the risk allele.

$\beta_{Flux,SNP_i}$ is flux effect size in individuals with dosage $i$ of the risk allele.

Flux is the reaction flux value. Adjusting flux values to regress out any potential effects of SNP dosage has no effect in this analysis as the dosage of the risk allele is constant for each test.

The CoxPHFitter function from the lifelines python package (Davidson-Pilon, 2019) was used to estimate the effect size of reaction flux values and its standard error (SE) for each dosage of the risk allele. A Welch's ANOVA test (i.e. a variant of ANOVA that does not assume homogeneity of variance) (Wilcox, 2003) was used to evaluate if there were significant differences between reaction effect sizes across risk allele dosages. To facilitate comparing these results with the interaction effect size, effect size variation per dosage of risk allele was computed with a linear regression of flux effect size per dosage weighted by the standard error of effect sizes estimates at each dosage ($1/SE^2$).

In both tests, the first ten genetic principal components were also used as covariates but have been omitted from the equations for clarity. In addition, this analysis was stratified by sex and in UKB by genotyping array. To test the effect of additional cardiometabolic factors, the two statistical models were also run using BMI, systolic blood pressure, LDL-cholesterol, HDL-cholesterol, and triglycerides, measured at the time of UKB enrolment, as covariates.

## Variant effects per flux quartile

To help visualize the amplification and buffering of risk alleles by reaction fluxes (Fig. 2), individuals were grouped into quartiles according to the reaction flux value for each analysed reaction and variant effect sizes were estimated within each quartile using a Cox proportional hazards model with age as the time scale.

$$h_{Flux_q}(t) = h0_{Flux_q}(t) \cdot \exp(\beta_{SNP,Flux_q} \cdot SNP_{Flux_q})$$

$h_{Flux_q}(t)$ is the hazard function defining the risk of developing CAD at age $t$ in individuals within quartile $q$ of the analysed reaction flux.

$h0_{Flux_q}$ is the baseline hazard in individuals within quartile $q$ of the analysed reaction flux.

$\beta_{SNP,Flux_q}$ is the variant effect size in individuals within quartile $q$ of the analysed reaction flux.

SNP is the risk allele dosage.

The CoxPHFitter (Davidson-Pilon, 2019) was used to estimate the variant effect size and SE using the first ten genetic principal components as covariates stratifying by sex and genotyping array. The resulting effect sizes and SE were used to visualize the effect of having high or low flux values in risk allele effect sizes (Fig. 2).

## Reaction selection and pruning

In the HUMAN1-derived organ-specific models (Robinson et al, 2020; Foguet et al, 2022), some reactions lack gene annotation. This can occur because they are spontaneous processes, but more often than not this arises because the gene(s) mediating the reaction are unknown or because they are artificial reactions needed to balance the models (e.g., exchange reactions). As such, we decided to focus the amplification/buffering analysis only on the fluxes of reactions

with annotated genes (adipose tissue: 721, brain: 1116, heart: 1314, liver: 1956, skeletal muscle: 1078).

In addition, in any metabolic network, many reaction fluxes have a significant degree of flux correlation. This can arise from stochiometric coupling between reactions (e.g., the product of the first reaction is the substrate of the second reaction) and from proteins that mediate multiple reactions (e.g. some transmembrane carriers can transport a wide range of substrates). To account for this and facilitate the interpretation of the results, for each analysed organ metabolic network, we pruned reaction fluxes with more than 50% correlation from the interaction results. Briefly, Pearson correlation coefficients were computed between all reaction fluxes from each organ. Next, all SNP-reaction fluxes pairs were ranked based on the maximum $P$-value between the interaction effect size test and the dosage-specific test. Then, starting from the reaction flux in the most significant reaction-SNP pair, reactions with more than 0.5 flux correlation (r) to this reaction were identified and all pairs involving these reactions were removed. The process was subsequently repeated for all ranked reaction-SNP pairs until no pairs involving reactions from the same organ-metabolic network with more than 0.5 flux correlation remained. In total 1670 reactions remained (adipose tissue: 280, brain: 417, heart: 360, liver: 263, skeletal muscle: 350).

The interaction effect size test and the dosage-specific test $P$-values were adjusted for multiple testing against all remaining SNP–reaction flux pairs using the Benjamini and Hochberg (i.e., FDR) method.

In addition, the univariate effect of the uncorrelated reactions on disease risk was also evaluated using a Cox model stratifying by sex and genotyping array and using the first 10 genetic principal components as covariates. The resulting $P$-values were FDR-adjusted for all uncorrelated reactions within each disease definition (Appendix Fig. S4).

## Linkage disequilibrium estimation and variant annotation

LD between risk SNPs and/or SNPs used in flux prediction was measured with the genotype data from the UKB subset of inferred European ancestry using Plink 1.9 (Chang et al, 2015). LD blocks used to delimitate independent risk loci were defined using hierarchical clustering with the single linkage method ("friends of friends" clustering) using $1-R^2$ between risk variants as a distance metric. The resulting hierarchical tree was cut at a height of 0.4, identifying LD blocks such that each risk SNP had an $R^2 > 0.6$ with at least one other risk SNP within the block and an $R^2 < 0.6$ with all risk SNPs outside the block.

Ensembl Variant Effect Predictor was used to annotate the effect of risk variants involved in interactions (McLaren et al, 2016). Similarly, known eQTL and sQTL variants and their target genes were obtained from GTEx Portal (https://gtexportal.org/home/) (GTEx Consortium, 2013) and the INTERVAL RNA-SEQ Portal (https://www.intervalrna.org.uk/) (Tokolyi et al, 2025) and were used to further annotate the risk variants involved in interactions.

Gene features shown in plots were extracted from the TxDb.Hsapiens.UCSC.hg19.knownGene package with the make-GenesDataFromTxDb function from the karyoploteR package (Gel and Serra, 2017).

## Data availability

Data from the UK Biobank and All of Us Research Program cohorts is under restricted access due to the presence of potentially sensitive individual-level data. Researchers can request access by registering and submitting a reasoned application to the UK Biobank (https://www.ukbiobank.ac.uk/) and All of Us Research Program (https://www.researchallofus.org/). The summary statistics for interaction effects are available in the Zenodo dataset repository (https://doi.org/10.5281/zenodo.14919939). In addition, interactions with a nominal *P*-value below 0.001 can be visualized and queried through interactive widgets hosted at https://cfoguet.github.io/datasets/.

The source data of this paper are collected in the following database record: biostudies:S-SCDT-10_1038-S44320-025-00097-2.

## Peer review information

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

## Acknowledgements

This research has been conducted using the UK Biobank Resource under Application 7439 and linked health data provided by patients and collected by the NHS as part of their care and support. This work was performed using

resources provided by the Cambridge Service for Data-Driven Discovery (CSD3) operated by the University of Cambridge Research Computing Service (www.csd3.cam.ac.uk), provided by Dell EMC and Intel using Tier-2 funding from the Engineering and Physical Sciences Research Council (capital grant EP/P020259/1), and DiRAC funding from the Science and Technology Facilities Council (www.dirac.ac.uk). This work was supported by core funding from the British Heart Foundation (RG/18/13/33946; RG/F/23/110103), NIHR Cambridge Biomedical Research Centre (NIHR203312) [*], BHF Chair Award (CH/12/2/29428), Cambridge BHF Centre of Research Excellence (RE/18/1/34212), and by Health Data Research UK, which is funded by the UK Medical Research Council, Engineering and Physical Sciences Research Council, Economic and Social Research Council, Department of Health and Social Care (England), Chief Scientist Office of the Scottish Government Health and Social Care Directorates, Health and Social Care Research and Development Division (Welsh Government), Public Health Agency (Northern Ireland), British Heart Foundation and the Wellcome Trust. XJ was also supported by the Wellcome Trust (227566/Z/23/Z). MI was also supported by the UK Economic and Social Research Council (ES/T013192/1). We gratefully acknowledge the National Institutes of Health's All of Us Research Program for making available the participant data examined in this study. We also thank All of Us and UKB participants for their contributions, without whom this research would not have been possible. *The views expressed are those of the authors and not necessarily those of the NIHR or the Department of Health and Social Care.

## Author contributions

**Carles Foguet**: Conceptualization; Data curation; Software; Formal analysis; Visualization; Methodology; Writing—original draft; Writing—review and editing. **Xilin Jiang**: Formal analysis; Methodology; Writing—review and editing. **Scott C Ritchie**: Data curation; Software; Methodology; Writing—review and editing. **Elodie Persyn**: Software; Visualization; Methodology; Writing—review and editing. **Yu Xu**: Software; Methodology; Writing—review and editing. **Chief Ben-Eghan**: Software; Methodology; Writing—review and editing. **Henry J Taylor**: Software; Validation; Methodology; Writing—review and editing. **Emanuele Di Angelantonio**: Resources; Funding acquisition; Project administration; Writing—review and editing. **John Danesh**: Resources; Funding acquisition; Project administration; Writing—review and editing. **Adam S Butterworth**: Resources; Funding acquisition; Project administration; Writing—review and editing. **Samuel A Lambert**: Resources; Funding acquisition; Methodology; Writing—review and editing. **Michael Inouye**: Conceptualization; Resources; Supervision; Funding acquisition; Methodology; Writing—original draft; Project administration; Writing—review and editing.

Source data underlying figure panels in this paper may have individual authorship assigned. Where available, figure panel/source data authorship is listed in the following database record: biostudies:S-SCDT-10_1038-S44320-025-00097-2.

## Disclosure and competing interests statement

ASB reports institutional grants from AstraZeneca, Bayer, Biogen, BioMarin, Bioverativ, Novartis, Regeneron and Sanofi. JD serves on scientific advisory boards for AstraZeneca, Novartis, Our Future Health and UK Biobank, and has received multiple grants from academic, charitable and industry sources outside of the submitted work. MI is a trustee of the Public Health Genomics (PHG) Foundation, a member of the Scientific Advisory Board of Open Targets, and has research collaborations with AstraZeneca, Nightingale Health and Pfizer which are unrelated to this study.

# Expanded View Figures

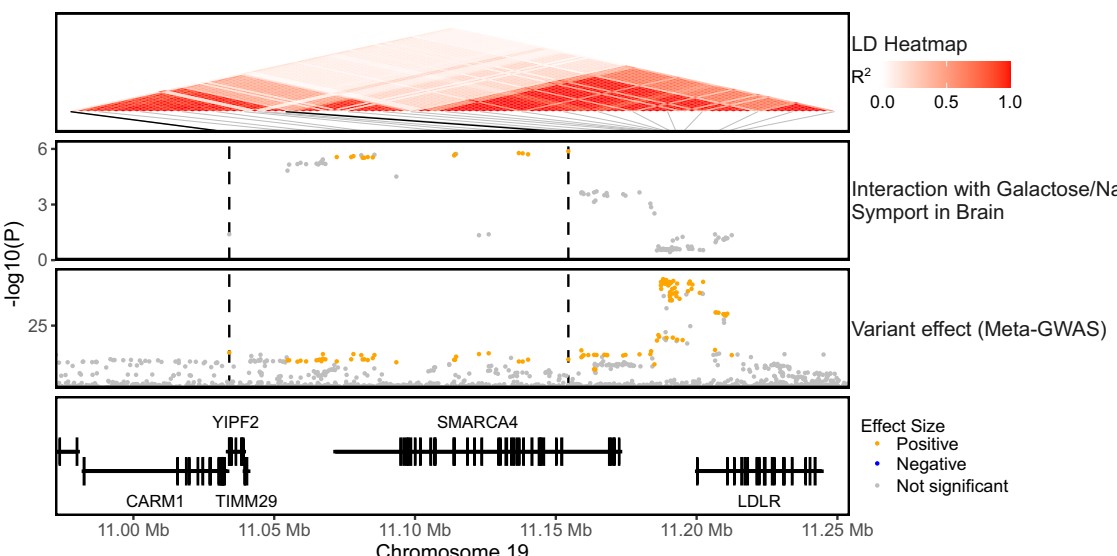

**Figure EV1.   Interaction between galactose transport in brain and variants in the *SMARCA4* risk locus.**

The regional association plots show the −log10(*P*-value) for interaction and variant effect sizes on CAD risk.

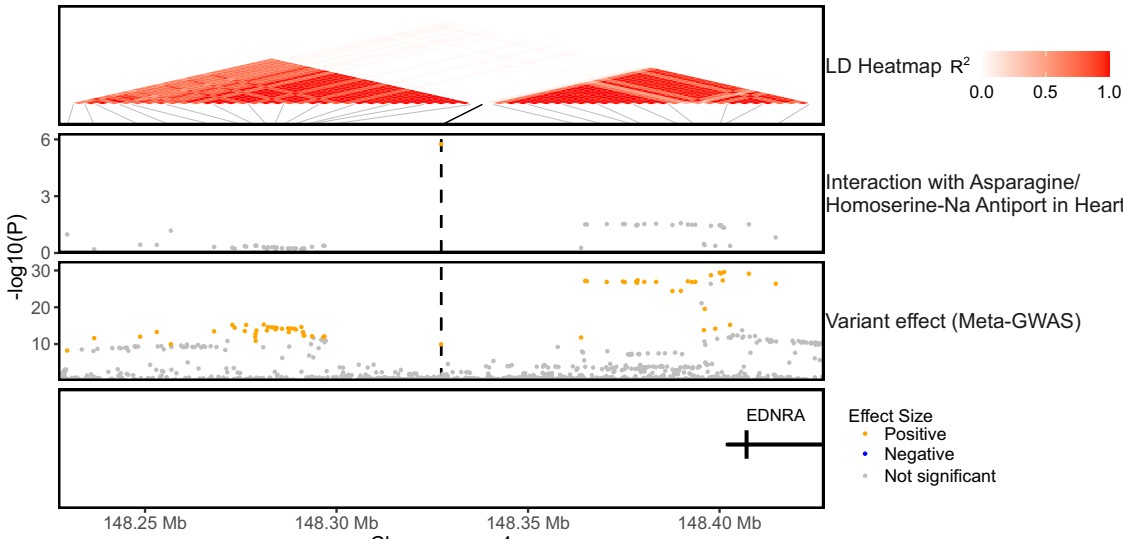

**Figure EV2. Amino acid transport amplifies the effect of a risk variant at the *EDNRA* locus.**

The regional association plots show the −log10(*P*-value) for interaction and variant effect sizes on CAD risk.

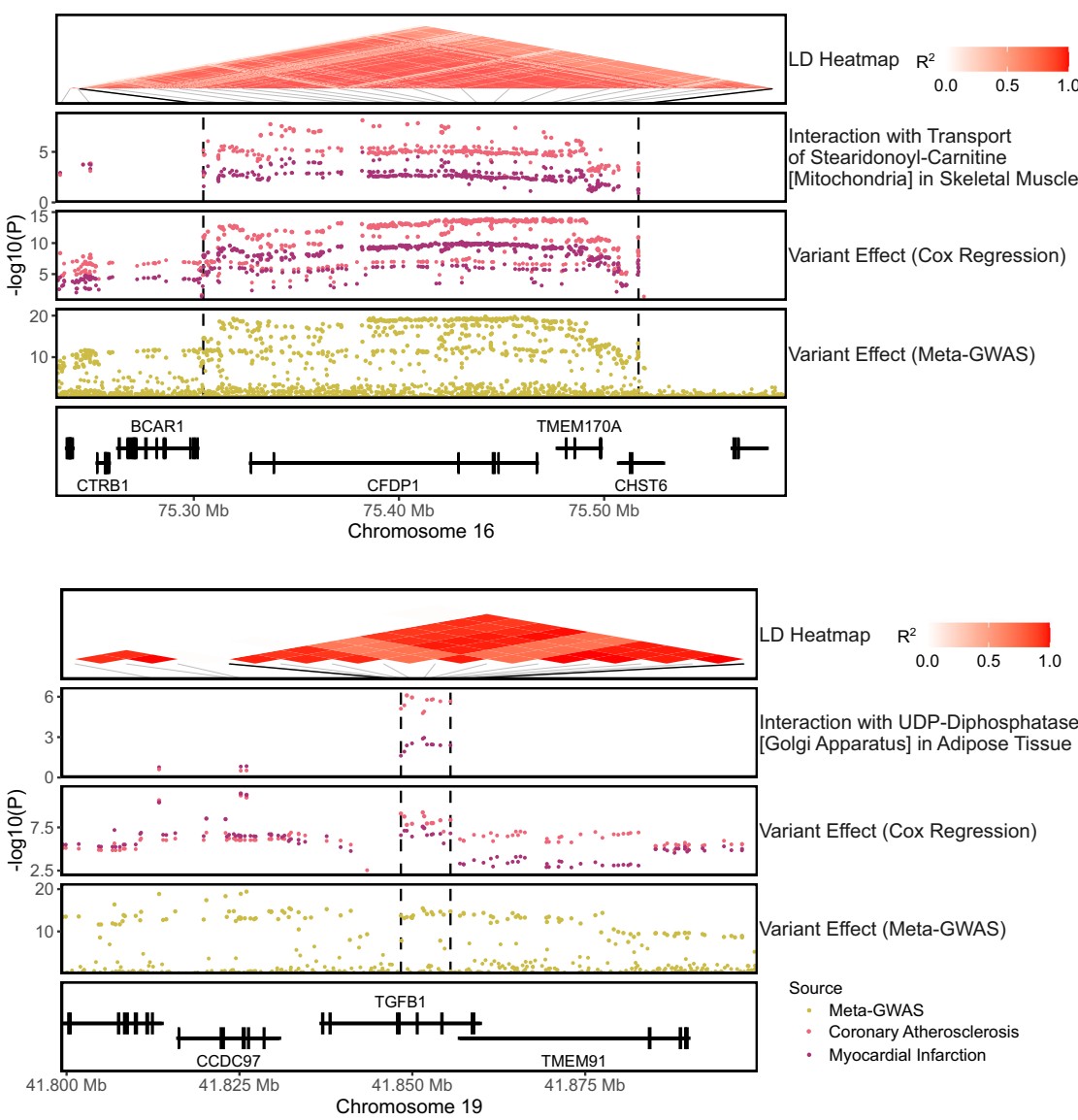

**Figure EV3. Example of interactions that are attenuated in myocardial infarction.**

The regional association plots show the −log10(P-value) for interaction and variant effect sizes on disease risk. P-values for interaction and variant effects (Cox Regression) were derived from the interaction and variant effect size tests, respectively, for coronary atherosclerosis or myocardial infarction events. As a reference, the variant effects from the meta-GWAS on CAD risk (Aragam et al, 2022) are also plotted.

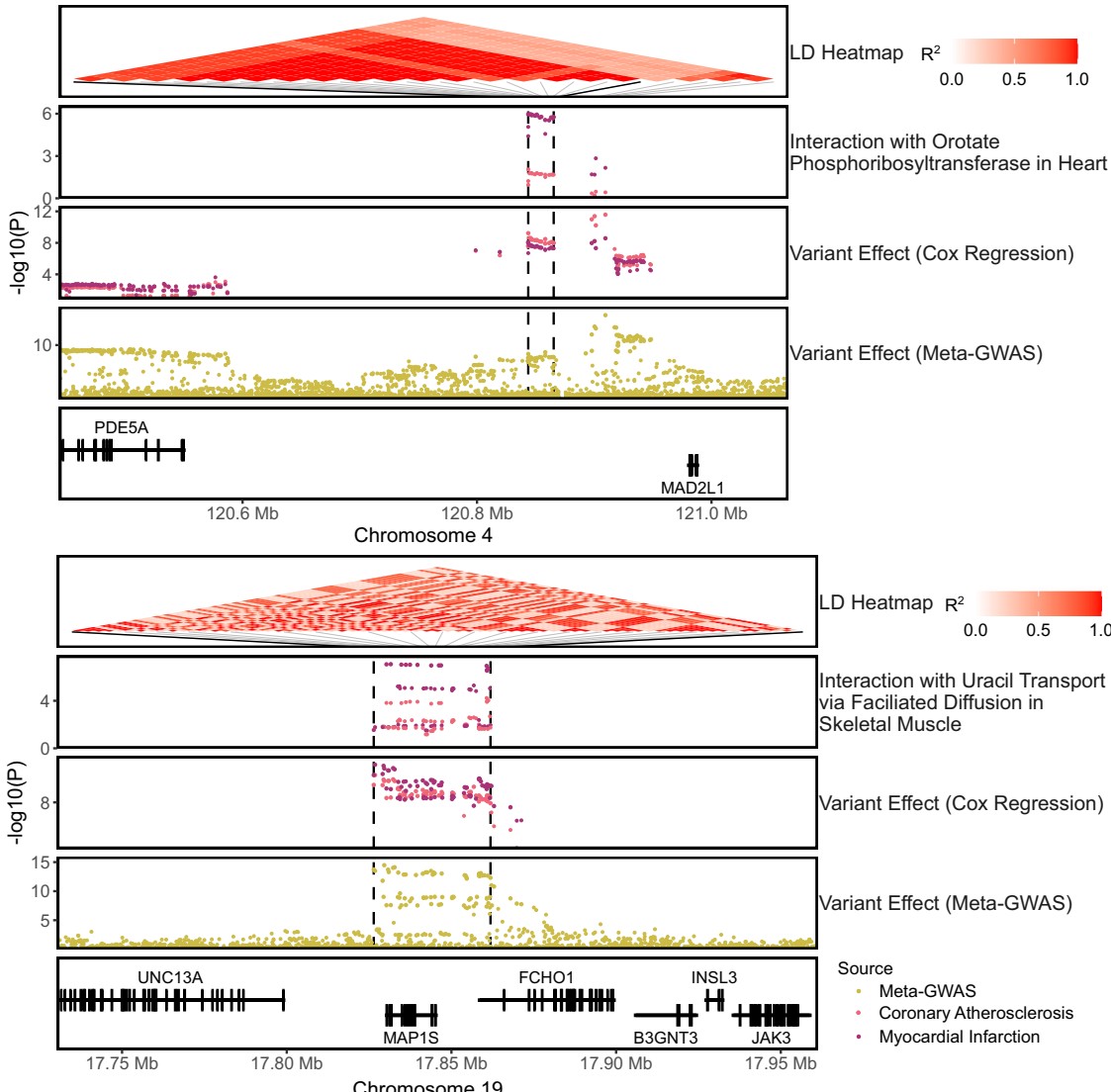

**Figure EV4. Example of interactions that are specific to myocardial infarction.**

The regional association plots show the −log10(*P*-value) for interaction and variant effect sizes on disease risk. *P*-values for interaction and variant effects (Cox Regression) were derived from the interaction and variant effect size tests, respectively, for coronary atherosclerosis or myocardial infarction events. As a reference, the variant effects from the meta-GWAS on CAD risk (Aragam et al, 2022) are also plotted.

