## [Peer Review File · Molecular Systems Biology]

Metabolic reaction fluxes as amplifiers and buffers of risk alleles for coronary artery disease

Carles Foguet, Xilin Jiang, Scott Ritchie, Elodie Persyn, Yu Xu, Chief Ben-Eghan, Henry Taylor, Emanuele Di Angelantonio, John Danesh, Adam Butterworth, Samuel Lambert, and Michael Inouye

Corresponding author(s): Carles Foguet (cf545@medschl.cam.ac.uk) , Michael Inouye (mi336@cam.ac.uk)

Review Timeline:

Submission Date:	4th Sep 24
Editorial Decision:	29th Oct 24
Revision Received:	6th Mar 25
Editorial Decision:	11th Mar 25
Revision Received:	12th Mar 25
Accepted:	13th Mar 25

Editor: Jingyi Hou

Transaction Report:

29th Oct 2024

Manuscript Number: MSB-2024-12609

Title: Metabolic reaction fluxes as amplifiers and buffers of risk alleles for coronary artery disease

Author: Carles Foguet

Xilin Jiang

Scott Ritchie

Elodie Persyn

Yu Xu

Chief Ben-Eghan

Emanuele Di Angelantonio

John Danesh

Adam Butterworth

Samuel Lambert

Michael Inouye

Dear Dr Foguet,

Thank you for submitting your work to Molecular Systems Biology. We have now heard back from the three reviewers who agreed to evaluate your manuscript. As you will see from the reports below, the reviewers acknowledge the interest of the study. They raise, however, a series of concerns, which we would ask you to address in a major revision.

The reviewers raised some overlapping concerns, including the absence of an independent cohort (noted by Reviewers #2 and #3) and a lack of validations (mentioned by Reviewers #1 and #3). We would ask you to discuss the limitation of not having an independent cohort (in case adding another cohort is not feasible) and to conduct some levels of experimental validation, as suggested by Reviewer #1. Additionally, Reviewer #1 recommended estimating phenotypic variation in the flux estimates, which should be carefully addressed, along with other issues raised.

As you may already know, our editorial policy allows in principle a single round of major revision, and it is therefore essential to provide responses to the reviewers' comments that are as complete as possible.

On a more editorial level, we would ask you to address the following issues:

- Please provide a .docx formatted version of the manuscript text (including legends for main figures, EV figures and tables). Please make sure that the changes are highlighted to be clearly visible.

- Please provide individual production quality figure files as .eps, .tif, .jpg (one file per figure).

-Please provide a .docx formatted letter INCLUDING the reviewers' reports and your detailed point-by-point responses to their comments. As part of the EMBO Press transparent editorial process, the point-by-point response is part of the Review Process File (RPF), which will be published alongside your paper.

-Please note that all corresponding authors are required to supply an ORCID ID for their name upon submission of a revised manuscript.

-We replaced Supplementary Information with Expanded View (EV) Figures and Tables that are collapsible/expandable online (see examples in <http://msb.embopress.org/content/11/6/812>). A maximum of 5 EV Figures can be typeset. EV Figures should be cited as 'Figure EV1, Figure EV2' etc... in the text and their respective legends should be included in the main text after the legends of regular figures.

Additional Tables/Datasets should be labeled and referred to as Table EV1, Dataset EV1, etc. Legends have to be provided in a separate tab in case of .xls files. Alternatively, the legend can be supplied as a separate text file (README) and zipped together with the Table/Dataset file.

For the figures and tables that you do NOT wish to display as Expanded View figures, they should be bundled together with their legends in a single PDF file called *Appendix*, which should start with a short Table of Content. Each legend should be below the corresponding Figure/Table in the Appendix. Appendix figures and tables should be referred to in the main text as:

"Appendix Figure S1, Appendix Figure S2, Appendix Table S1" etc. See detailed instructions regarding expanded view here: <https://www.embopress.org/page/journal/17444292/authorguide#expandedview>.

-Before submitting your revision, primary datasets (and computer code, where appropriate) produced in this study need to be deposited in an appropriate public database (see <http://msb.embopress.org/authorguide> - dataavailability <https://www.embopress.org/page/journal/17444292/authorguide#dataavailability>).

The accession numbers and database should be listed in a formal "Data Availability" section (placed after Materials & Method) that follows the model below (see also <https://www.embopress.org/page/journal/17444292/authorguide#dataavailability>). Please note that the Data Availability Section is restricted to new primary data that are part of this study.

Data availability

-At EMBO Press we ask authors to provide source data for the main figures. Our source data coordinator will contact you to discuss which figure panels we would need source data for and will also provide you with helpful tips on how to upload and organize the files.

- Our journal encourages inclusion of *data citations in the reference list* to directly cite datasets that were re-used and obtained from public databases. Data citations in the article text are distinct from normal bibliographical citations and should directly link to the database records from which the data can be accessed. In the main text, data citations are formatted as follows: "Data ref: Smith et al, 2001". In the Reference list, data citations must be labeled with "[DATASET]". A data reference must provide the database name, accession number/identifiers and a resolvable link to the landing page from which the data can be accessed at the end of the reference. Further instructions are available at .

- We updated our journal's competing interests policy in January 2022 and request authors to consider both actual and perceived competing interests. Please review the policy <https://www.embopress.org/competing-interests> and update your competing interests if necessary.

Please use the heading "Disclosure statement and competing interests".

- All Materials and Methods need to be described in the main text using our 'Structured Methods' format. According to this format, the Methods section includes a Reagents and Tools Table (listing key reagents, experimental models, software and relevant equipment and including their sources and relevant identifiers) followed by a Methods and Protocols section describing the methods, ideally using a step-by-step protocol format. The aim is to facilitate adoption of the methodologies across labs. Please download and fill our Reagents and Tools Table template (.docx), which you can find in our author guidelines:

<https://www.embopress.org/page/journal/17444292/authorguide#structuredmethods>.

-Regarding data quantification:

Please ensure to specify the name of the statistical test used to generate error bars and P values, the number (n) of independent experiments (please specify technical or biological replicates) underlying each data point and the test used to calculate p-values in each figure legend. Discussion of statistical methodology can be reported in the materials and methods section, but figure legends should contain a basic description of n, P and the test applied.

Graphs must include a description of the bars and the error bars (s.d., s.e.m.).

- Please provide a "standfirst text" summarizing the study in one or two sentences (approximately 250 characters, including space), three to four "bullet points" highlighting the main findings and a "synopsis image" (550px width and 400-600 px height, PNG format) to highlight the paper on our homepage.

Here are a couple of examples:

<https://www.embopress.org/doi/10.15252/msb.20199356>

<https://www.embopress.org/doi/10.15252/msb.20209475>

<https://www.embopress.org/doi/10.15252/msb.209495>

When you resubmit your manuscript, please download our CHECKLIST (<https://www.embopress.org/pb-assets/embo-site/EMBO%20Press%20Author%20Checklist-1642513524327.xlsx>) and include the completed form in your submission.

Please note that the Author Checklist will be published alongside the paper as part of the transparent process (<https://www.embopress.org/page/journal/17444292/authorguide#transparentprocess>).

If you feel you can satisfactorily deal with these points and those listed by the referees, you may wish to submit a revised version of your manuscript. Please attach a covering letter giving details of the way in which you have handled each of the points raised by the referees. A revised manuscript will be once again subject to review and you probably understand that we can give you no guarantee at this stage that the eventual outcome will be favorable.

When you resubmit your manuscript, please download our CHECKLIST (<https://bit.ly/EMBOPressAuthorChecklist>) and include the completed form in your submission.

Please note that the Author Checklist will be published alongside the paper as part of the transparent process (<https://www.embopress.org/page/journal/17444292/authorguide#transparentprocess>).

If you feel you can satisfactorily deal with these points and those listed by the referees, you may wish to submit a revised version of your manuscript. Please attach a covering letter giving details of the way in which you have handled each of the points raised by the referees. A revised manuscript will be once again subject to review and you probably understand that we can give you no guarantee at this stage that the eventual outcome will be favorable.

I look forward to receiving your revised manuscript soon.

Kind regards,
Jingyi

Jingyi Hou, PhD
Scientific Editor
Molecular Systems Biology

We realize that it is difficult to revise to a specific deadline. In the interest of protecting the conceptual advance provided by the work, we recommend a revision within 3 months (27th Jan 2025). Please discuss the revision progress ahead of this time with the editor if you require more time to complete the revisions. Use the link below to submit your revision:

IMPORTANT: When you send your revision, we will require the following items:

1. the manuscript text in LaTeX, RTF or MS Word format
2. a letter with a detailed description of the changes made in response to the referees. Please specify clearly the exact places in the text (pages and paragraphs) where each change has been made in response to each specific comment given
3. three to four 'bullet points' highlighting the main findings of your study
4. a short 'blurb' text summarizing in two sentences the study (max. 250 characters)
5. a 'thumbnail image' (550px width and max 400px height, Illustrator, PowerPoint or jpeg format), which can be used as 'visual title' for the synopsis section of your paper.
6. Please include an author contributions statement after the Acknowledgements section (see <https://www.embopress.org/page/journal/17444292/authorguide>)
7. Please complete the CHECKLIST available at (<https://bit.ly/EMBOPressAuthorChecklist>).

Please note that the Author Checklist will be published alongside the paper as part of the transparent process (<https://www.embopress.org/page/journal/17444292/authorguide#transparentprocess>).

See also figure legend guidelines: <https://www.embopress.org/page/journal/17444292/authorguide#figureformat>

9. Please note that corresponding authors are required to supply an ORCID ID for their name upon submission of a revised manuscript (EMBO Press signed a joint statement to encourage ORCID adoption).

(<https://www.embopress.org/page/journal/17444292/authorguide#editorialprocess>)

Currently, our records indicate that the ORCID for your account is 0000-0001-8494-9595.

Link Not Available

11. Include a Reagents and Tools Table as part of the Methods section, which can be downloaded from our author guidelines (<https://www.embopress.org/page/journal/17444292/authorguide#structuredmethods>)

*** PLEASE NOTE *** As part of the EMBO Press transparent editorial process initiative (see our Editorial at <https://dx.doi.org/10.1038/msb.2010.72>), Molecular Systems Biology publishes online a Review Process File with each accepted

manuscripts. This file will be published in conjunction with your paper and will include the anonymous referee reports, your point-by-point response and all pertinent correspondence relating to the manuscript. If you do NOT want this File to be published, please inform the editorial office at msb@embo.org within 14 days upon receipt of the present letter.

Reviewer #1:

This is an overall elegant paper that highlights the potential of integrating GWAS data (specifically in CAD domain, but it can be generally applicable) with predicted flux data from genome-scale metabolic models (GEMs) - ultimately resulting in insights how the genetic variation impacts host metabolism at the flux level. This is potentially a powerful approach, but it seems some shortcuts had to be understandably taken that are not sufficiently discussed as limitations. While choice of metabolic reconstruction (HUMAN1) is reasonable, as a general issue with the approach the flux estimates are not truly personalised, except at the SNP level. The approach does not capture large variation in fluxes as a result of phenotypical variation (BMI, sex, liver fat, etc). Given the sample sizes, this may be understandable but it does raise question how reliable the flux estimates truly are. The manuscript would be strengthened by (1) estimate of phenotypic variation on estimated fluxes, and (2) some experimental validation of observed interaction of genetic variation and fluxes, even if in vitro.

While the manuscript is overall well written, the presentation of the data is sometimes difficult to follow. References to figures do not always follow the numeric order, and some figures need improvement (particularly legends to figs 2 and 3 should be made clearer).

Reviewer #2:

Foguet and colleagues present a tour-de-force analysis in which they leverage UK BioBank data in two orthogonal ways. First, they use existing organ-specific metabolic modeling algorithms to develop genetically personalized models for metabolic pathways, and from these to predict genotype-specific reaction fluxes across grouped pathways. Separately, they interrogate the UKBB electronic health records to bin individuals into either those with evidence of coronary artery disease (CAD) or controls with no evidence of CAD in the records. By comparing the CAD group to controls the researchers filter the totality of the genetic variation down to ~6k SNPs that most strongly differentiate CAD from control. They then looked for statistical interaction between these SNPs and the predicted reaction fluxes derived separately, to identify pathway-flux level modifiers of CAD risk. In so doing, they observed an impressively high fraction (530/583) of the risk-allele/flux pairings map to a locus on chromosome 6 that harbors both the LPA and PLG genes, encoding apolipoprotein(a) and plasminogen respectively. They show evidence of predicted interaction between SNPs at this locus with predicted reaction fluxes that would be expected to impact levels of bioactive metabolites including polyamines in muscle and inflammatory mediators including histamine prostaglandins.

Overall, this is an impressive use of cutting edge metabologenomic study that takes advantage of a large electronic health biobank database, sophisticated statistical modeling, and the approach is validated by a number of convincing vignettes from the data. Conceptually, this is reminiscent of Mendelian randomization, but across the physiome. While the choice of CAD as a disease endpoint is very well justified, one can imagine that this approach could equally well be applied to identifying pathway-level modifiers of disease for situations where the etiology is less clear, and thereby potential identifying novel therapeutic axes. As such, this work is of broad potential interest.

I'm generally supportive of accepting the paper as is, so the remaining remarks are relatively minor:

The figure quality is not great in the review pdf - the SNP symbols are tiny in the final assembled image - suggest making them bigger. For Figures 3-5 the LD heat maps between the locus plots and gene map is visually confusing - suggest inverting the LD plot and putting it above the top locus plot so that the gene map is directly below the bottommost plot (Meta-GWAS).

The authors explicitly include a vignette covering carnitine flux in Figure 4, but while Figure 3 includes SLC22A1 - which encodes OCT1, the transporter that regulates acyl-carnitine flux between the liver and other tissues, there is no plot referring to carnitine flux for Figure 3 - are there no effects that survive statistical correction? Fine mapping of the SLC22A1 locus has shown that there are two separate functional SNPs that impact carnitine levels, and that these are on different LD blocks (the authors' LD plot includes the LD break within SLC22A1. Does the current analysis miss edge cases like this?

For the daunting amount of computational work involved in the paper, I'm surprised to not see an associated publicly available resource - a web tool or shiny app. The ability to interrogate selected loci with specific hypotheses would be invaluable to for other groups involved in efforts to do functional follow up studies on GWAS loci without clear links to known biology - even if those loci fail to pass nominal statistical significance correction for multiple testing. How do the authors justify not making the work product more accessible?

The only significant concern about the present study is the lack of replication an independent cohort, which somewhat limits the confidence with which the results can be interpreted.

Reviewer #3:

Foguet and colleagues have presented a study examining the non-additive effects of biochemical reaction fluxes with CAD risk alleles on CAD susceptibility. While the authors provide a reasonable study design and significant findings, I believe the manuscript would benefit from clarifying the following points in the manuscript to enhance readability:

1. Why was a Cox regression model chosen instead of a regression on a binary outcome?
2. Have the authors attempted to replicate the tests in another cohort?
3. The authors proposed potential mechanisms for the interaction between risk alleles and metabolic pathways. How could these proposed mechanisms be validated?
4. The authors mentioned that some eQTLs used to estimate flux values are not in strong LD with the CAD risk alleles. What about those in strong LD? Could the authors discuss these as well?

Reviewer 1

This is an overall elegant paper that highlights the potential of integrating GWAS data (specifically in CAD domain, but it can be generally applicable) with predicted flux data from genome-scale metabolic models (GEMs) - ultimately resulting in insights how the genetic variation impacts host metabolism at the flux level. This is potentially a powerful approach, but it seems some shortcuts had to be understandably taken that are not sufficiently discussed as limitations. While choice of metabolic reconstruction (HUMAN1) is reasonable, as a general issue with the approach the flux estimates are not truly personalised, except at the SNP level. The approach does not capture large variation in fluxes as a result of phenotypical variation (BMI, sex, liver fat, etc). Given the sample sizes, this may be understandable but it does raise question how reliable the flux estimates truly are. The manuscript would be strengthened by (1) estimate of phenotypic variation on estimated fluxes, and (2) some experimental validation of observed interaction of genetic variation and fluxes, even if in vitro.

We thank the reviewer for their positive feedback. Following their advice, we now explicitly discuss the limitations of using fluxes simulated from genotype data. To predict metabolic fluxes, we require organ-specific transcript abundances and direct measures for those are not available in large prospective cohorts suitable for disease risk association studies such as UK Biobank or All of Us Research Program. Thus, as the reviewer rightly notes, using genetically imputed measures is necessary to leverage the large sample sizes arising from the large number of genotyped individuals in these cohorts. For this reason, genotype-derived measures are frequently used as a proxy to identify molecular features associated with disease risk (Xu et al. 2023; Lu et al. 2023; Wingo et al. 2021).

As noted, genetically predicted fluxes do not capture flux variation arising from environmental factors. However, this is also advantageous as it minimises the risk of confounding effects that could lead to spurious associations on CAD risk. It is worth highlighting that our statistical model is already stratified by sex, hence sex-specific effects on fluxes would have no bearing on interaction estimates.

To confirm that our interaction effects were robust to phenotypic variation in other cardiometabolic traits, we tested the impact of also accounting for non-genetic factors. Namely, BMI, systolic blood pressure, and LDL-cholesterol, HDL-cholesterol, and triglycerides were added as covariates to the statistical models we used to detect interaction. The inclusion of these additional cardiometabolic measures did not significantly affect the interaction effect estimates between reaction fluxes and risk variant dosage on CAD and MI risk, with correlation coefficients (r) exceeding 0.99 when compared to the statistical models without additional cardiometabolic covariates. This analysis is now described in the "Overview of the methods" section and is supported by a new supplementary figure:

Line 121: (In our analysis) we controlled for age, sex, and genetic principal components. We also evaluated the impact of including additional non-genetic cardiovascular risk factors as covariates, namely BMI, systolic blood pressure, and blood levels of LDL-cholesterol, HDL-cholesterol, and triglycerides

(Methods) finding that their inclusion did not significantly alter interaction effect estimates (Appendix Figure S2)

Appendix Figure S2: Robustness of interaction estimates to the inclusion of additional cardiometabolic covariates. Pairs of SNPs and reaction fluxes with significant interaction on coronary atherosclerosis or myocardial infarction risk were re-evaluated for interaction accounting for additional cardiometabolic measures (i.e., BMI, systolic blood pressure, and blood levels of LDL-cholesterol, HDL-cholesterol, and triglycerides) as covariates. Interaction effects were quantified using both the interaction effect size test and the dosage-specific test (Methods). The dashed line indicates the linear regression of interaction effects estimates with the additional covariates relative to the original estimates.

While we are supportive of experimental validation in general, it is unfortunately not currently feasible to test reaction flux-SNP interactions on CAD risk in experimental models. *In vitro* and *ex vivo* models of the disease would not capture the impact of organ-specific fluxes on atherosclerosis. Similarly, the interactions would be challenging to validate in animal models due to different genetic architecture (e.g. lack of functionally equivalent SNPs to the ones amplified or buffered in humans). In this

regard, the variant-to-function (V2F) gap also makes it difficult to establish the precise molecular mechanisms of action for non-coding variants and would further difficult experimental validation. As an alternative, notably common in the field of human genetics, we have externally validated the interactions by showing their replication in an external cohort (All of Us Research Program).

We have also added a limitation paragraph extensively outlining the lack of experimental validation and the fact that our fluxes are genetically predicted as limitation:

Line 417: Our study has limitations. First, the necessary use of fluxes simulated from genetically imputed transcript abundances(Foguet et al, 2022) cannot account for the influence of environmental factors on reaction activities. However, this is also advantageous as it minimises confounding by environment-driven factors (e.g. diet, lifestyle, or BMI) on reaction fluxes and disease risk associations. Notably, we found that the detected interactions were not attenuated if direct cardiometabolic measures were included in the analysis. Furthermore, using genetically predicted fluxes enabled us to tap into the large sample sizes arising from the hundreds of thousands of genotyped individuals in large prospective cohorts such as the UKB and AoUs. Second, the genetically imputed fluxes used in this analysis do not explicitly account for the effect of coding variants and splicing variants. There is a lack of systematic quantitative annotation on their impact on enzyme activities which presently limits our capacity to incorporate their effects on our metabolic flux modelling framework(Foguet et al, 2022) but hopefully will enhance it in the future. Third, the experimental validation of reaction flux amplification/buffer on human alleles is not feasible. Such validation would face significant challenges and would need to be tackled on a case-by-case basis as the variant-to-function (V2F) gap makes it difficult to establish the precise molecular mechanisms of action for non-coding variants (Nandakumar et al, 2020; Claussnitzer & Susztak, 2021). Nevertheless, based on existing literature and established biological pathways, we propose mechanisms that could serve as the starting point for future research.

While the manuscript is overall well written, the presentation of the data is sometimes difficult to follow. References to figures do not always follow the numeric order, and some figures need improvement (particularly legends to figs 2 and 3 should be made clearer).

We thank the reviewer for the feedback. We have simplified figure legends and made the figures easier to read by using a larger font size and simplifying axis labels. Additionally, we have also removed redundant references to figures and tables that were not necessary to understand the results.

Reviewer 2

Foguet and colleagues present a tour-de-force analysis in which they leverage UK BioBank data in two orthogonal ways. First, they use existing organ-specific metabolic modeling algorithms to develop genetically personalized models for metabolic pathways, and from these to predict genotype-specific reaction fluxes across grouped pathways. Separately, they interrogate the UKBB electronic health records to bin individuals into either those with

evidence of coronary artery disease (CAD) or controls with no evidence of CAD in the records. By comparing the CAD group to controls the researchers filter the totality of the genetic variation down to ~6k SNPs that most strongly differentiate CAD from control. They then looked for statistical interaction between these SNPs and the predicted reaction fluxes derived separately, to identify pathway-flux level modifiers of CAD risk. In so doing, they observed an impressively high fraction (530/583) of the risk-allele/flux pairings map to a locus on chromosome 6 that harbors both the LPA and PLG genes, encoding apolipoprotein(a) and plasminogen respectively. They show evidence of predicted interaction between SNPs at this locus with predicted reaction fluxes that would be expected to impact levels of bioactive metabolites including polyamines in muscle and inflammatory mediators including histamine prostaglandins.

Overall, this is an impressive use of cutting edge metabologenomic study that takes advantage of a large electronic health biobank database, sophisticated statistical modeling, and the approach is validated by a number of convincing vignettes from the data. Conceptually, this is reminiscent of Mendelian randomization, but across the physiome. While the choice of CAD as a disease endpoint is very well justified, one can imagine that this approach could equally well be applied to identifying pathway-level modifiers of disease for situations where the etiology is less clear, and thereby potential identifying novel therapeutic axes. As such, this work is of broad potential interest.

I'm generally supportive of accepting the paper as is, so the remaining remarks are relatively minor:

The figure quality is not great in the review pdf - the SNP symbols are tiny in the final assembled image - suggest making them bigger. For Figures 3-5 the LD heat maps between the locus plots and gene map is visually confusing - suggest inverting the LD plot and putting it above the top locus plot so that the gene map is directly below the bottommost plot (Meta-GWAS).

We are grateful for the kind words from the reviewer. We apologize for the quality issue with the figures in the PDF that was assessed; this was not an issue in our local files. To address this, we have enhanced our files through clearer and larger fonts to hopefully improve readability.

We have also moved the LD plot to the top of the regional plot. We agree with the reviewer that this makes the plot easier to interpret. For example:

The authors explicitly include a vignette covering carnitine flux in Figure 4, but while Figure 3 includes SLC22A1 - which encodes OCT1, the transporter that regulates acyl-carnitine flux between the liver and other tissues, there is no plot referring to carnitine flux for Figure 3 - are there no effects that survive statistical correction? Fine mapping of the SLC22A1 locus has shown that there are two separate functional SNPs that impact carnitine levels, and that these are on different LD blocks (the authors' LD plot includes the LD break within SLC22A1. Does the current analysis miss edge cases like this?

Overall, the vignettes in the paper highlight instances where we have identified amplification or buffering of risk alleles by reaction fluxes. The transport of stearidonoyl-carnitine to the mitochondria is the focus of Figure 4 and shows significant interactions between this transport process and a cluster of variants in chromosome 16. This transport process is mediated by SLC25A20 as defined by the gene reaction rules of the HUMAN1 reconstruction of human metabolism (Robinson et al. 2020). Other than the above, we detected no significant interactions involving acylcarnitine transport processes.

Regarding SLC22A1 mediated transport of acylcarnitines, our modelling framework does not presently account for the effect of alternative splicing or coding variants on transporter activity and function. This is primarily due to a lack of a systematic quantitative annotation of the effects of these variants on reaction activities. As such our simulated fluxes do not directly account for the effect of some of the coding SNPs of SLC22A1 which are known to modulate its function and acylcarnitine levels (Kim et al. 2017). We have added a sentence in the discussion outlining this as a limitation.

Line 425: Second, the genetically imputed fluxes used in this analysis do not explicitly account for the effect of coding variants and splicing variants. There is a lack of systematic quantitative annotation on their impact on enzyme activities which presently limits our capacity to incorporate their effects on our metabolic flux modelling framework(Foguet et al. 2022) but hopefully will enhance it in the future.

However, in our previous work (Foguet et al. 2022), using the same set of genetically personalized fluxes, we found several reaction fluxes significantly associated with the levels of acylcarnitines in the blood (including fluxes through acylcarnitine transport processes

mediated by SLC22A1). Therefore, even without accounting for coding and alternative splicing genetic variants, our flux model does encapsulate the role of SLC22A1 in influencing the acylcarnitine fluxes and their levels in the blood.

It is worth noting that the emphasis of this work is risk variant – reaction interaction on CAD risk but lack of interaction does not mean that a reaction or transport process is not biologically relevant. Instead, the lack of significant interactions involving the SLC22A1-mediated transport of acylcarnitine likely reflects that these reactions are not involved in interactions with the evaluated risk alleles for CAD risk or that if such interactions exist, it is not strong enough to meet the threshold for significance.

For the daunting amount of computational work involved in the paper, I'm surprised to not see an associated publicly available resource - a web tool or shiny app. The ability to interrogate selected loci with specific hypotheses would be invaluable to for other groups involved in efforts to do functional follow up studies on GWAS loci without clear links to known biology - even if those loci fail to pass nominal statistical significance correction for multiple testing. How do the authors justify not making the work product more accessible?

We appreciate the reviewer highlighting the immense amount of computational / analytical work that has already gone into the study. To extent this further, following the reviewer's advice, we have made interactive widgets to visualize and query those interactions with some nominal degree of significance ($P < 1e-3$):

- Overview of Summary Statistics for Coronary Atherosclerosis Risk
- Overview of Summary Statistics for Myocardial Infarction Risk

We have also made the full summary statistics available in Zenodo (<https://zenodo.org/records/14919940>). These resources are described in the data availability section.

The only significant concern about the present study is the lack of replication an independent cohort, which somewhat limits the confidence with which the results can be interpreted.

We appreciate the concern from the reviewer. We have now replicated the discovered interactions in the All of Us Program. While this cohort had significantly less power than UKB due to lower sample size and number of disease events, we found that many still surpassed multiple-testing adjustment and overall there was strong consistency between the interaction effect estimates in UKB and All of Us.

Line 330: External Validation in the All of Us cohort

To assess the reproducibility of the buffering or amplification effects identified in the UKB cohort, we conducted an external validation using data from the All of Us (AoUs) cohort (All of Us Research Program Investigators et al, 2019; All of Us Research Program Genomics Investigators, 2024). Genetically personalized fluxes were computed for all AoUs participants of European genetic ancestries (N=118,058). From the linked electronic health records, we identified 14,117 CAD cases and 75,204 non-CAD controls using the PheWAS Catalog definition of coronary atherosclerosis (Wu et

al, 2019). Then, the 583 SNP-reaction fluxes pairs with amplification or buffering effects in CAD risk discovered in UKB were evaluated for interaction effects in AoUs using both the interaction effect size test and the dosage-specific test (Methods). Despite a smaller sample size and fewer CAD cases in AoUs compared to UKB, interaction effect estimates were highly consistent between both cohorts ($r=0.796$ for the interaction effect size test and $r=0.80$ for the dosage-specific test; Figure 6 A-B). Out of the 583 pairs discovered in UKB, 253 were significant (FDR adjusted P -value <0.05) in AoUs with either the interaction effect size or dosage-specific test with an additional 131 pairs being borderline significant (FDR adjusted P -value <0.25 ; Appendix Dataset S1). Furthermore, 548 pairs of the 583 tested pairs (94%) showed consistent interaction effect size directions between UKB and AoUs.

Following the same approach, the 426 instances of amplification and buffering of MI risk discovered in UKB were also evaluated in AoUs (7,218 MI cases and 86,055 non-MI controls). Consistent with a lower phenotype heterogeneity between cohorts, the resulting interaction estimates for MI showed even greater consistency than those for CAD (MI: $r=0.9$ for the interaction effect sizes test and $r=0.91$ for the dosage-specific test; Figure 6 C-D). A total of 168 SNP-reaction fluxes pairs showed significant interactions in AoUs with either the interaction effect size or dosage-specific test and 175 additional pairs were borderline significant (Appendix Dataset S2). Furthermore, a total of 410 (96%) SNP-reaction fluxes pairs had consistent directionality of interaction effects estimates with UKB.

Figure 6. Validation of Interaction effects in the All of Us (AoUs) cohort. The pairs of SNP-reaction flux with significant interaction in coronary atherosclerosis (CAD) or myocardial infarction (MI) risk identified UKB were evaluated in the AoUs cohort using two complementary methods. In the first method, interaction is measured as the coefficient of the interaction term between the risk allele and reaction flux in CAD (A) or MI risk (C). The second method quantifies interactions as the variation of reaction flux effect sizes in CAD (B) or MI (D) risk across risk allele dosages (Methods). The dashed line indicates the linear regression of interaction effects between UKB and AoUs. Interaction effects are expressed as the $\log(\text{Hazard Ratio})$ per standard deviation of $\log(\text{flux value})$ and allele dosage on CAD or MI risk.

Reviewer 3

Foguet and colleagues have presented a study examining the non-additive effects of biochemical reaction fluxes with CAD risk alleles on CAD susceptibility. While the authors provide a reasonable study design and significant findings, I believe the manuscript would benefit from clarifying the following points in the manuscript to enhance readability:

1. Why was a Cox regression model chosen instead of a regression on a binary outcome?

We are grateful for the feedback. Cox regression models were chosen because they offer more statistical power in prospective studies where the time to disease event (in our case, using age as a time scale) is available (van der Net et al. 2008). Given that we anticipated that significant interactions could be hard to identify, statistical power was important for our study. We have previously demonstrated the validity of Cox regression models in to test for associations between genetically predicted fluxes (Foguet et al. 2022) or genetically predicted molecular features (Xu et al. 2023) and disease risk.

2. Have the authors attempted to replicate the tests in another cohort?

As part of this revision, we have replicated interactions between reaction fluxes and CAD risk allele in the All of Us Program (Line 330). A more detailed overview of the replication is provided in our reply to Reviewer 2's last comment.

3. The authors proposed potential mechanisms for the interaction between risk alleles and metabolic pathways. How could these proposed mechanisms be validated?

Experimental validation of the interaction between risk allele dosage and reaction fluxes on CAD susceptibility is unfortunately not currently feasible. We have previously addressed experimental validation in response to Reviewer 1 above. In summary, such validation would entail extensive experimental characterization of the mechanism of action by which the involved variant and reaction activities can modulate disease susceptibility and progression and would face significant challenges with either cell models or animal models. We now explicitly acknowledge this as a limitation in the discussion section (Line 429) .

4. The authors mentioned that some eQTLs used to estimate flux values are not in strong LD with the CAD risk alleles. What about those in strong LD? Could the authors discuss these as well?

When testing for interaction between reaction fluxes and CAD risk alleles, a potential cause of concern was instances where the risk allele would be with strong LD with an eQTL variant that had a strong individual effect on the flux value. This could result in a correlation between the risk allele dosage and flux value that could lead to spurious interactions. To minimise the risk of this, the effect of risk alleles on reaction fluxes was regressed out when testing for interactions (Methods). This approach was particularly effective because in the pairs of SNP-reaction flux with significant interactions there was no instance where an eQTL was both in strong LD with the CAD risk variant and had a strong individual effect on the reaction flux value.

We now realize that this point may not be clear in the original manuscript so we have reworked the paragraph and moved it to the section “Buffering and amplification of the effect of risk variants by reaction fluxes”

(Line: 151) We quantified the linkage disequilibrium (LD) between the 279 SNPs with significant interaction and the expression quantitative trait loci (eQTL) variants for metabolic gene expression used in flux computation. There were some instances where there was a strong LD between risk variants and metabolic eQTLs. This was expected given that metabolic genes are mapped to some of the CAD risk loci. However, if these eQTLs also had strong individual effects on reaction fluxes, they could introduce spurious interactions by correlating genetically predicted fluxes with risk allele dosage. To account for this, we had regressed the effect of risk alleles from reaction fluxes when testing for interactions (Methods). To validate this approach, we assessed the effect of individual eQTLs on the 18 reaction fluxes involved in significant interactions. None of the eQTLs in strong LD with CAD risk variants showed a strong correlation with the reaction fluxes identified as amplifiers or buffers of these variants' effects (Appendix Figure S5).

References

- Foguet, Carles, Yu Xu, Scott C Ritchie, Samuel A Lambert, Elodie Persyn, Artika P Nath, Emma E Davenport, et al. 2022. “Genetically Personalised Organ-Specific Metabolic Models in Health and Disease.” *Nature Communications* 13 (1): 7356. <https://doi.org/10.1038/s41467-022-35017-7>.
- Lin, Shiqi, Xingjian Gao, Frauke Degenhardt, Yu Qian, Tianzi Liu, Xavier Farre Ramon, Syed Sibte Hadi, et al. 2023. “Genome-Wide Epistasis Study Highlights Genetic Interactions Influencing Severity of COVID-19.” *European Journal of Epidemiology* 38 (8): 883–89. <https://doi.org/10.1007/s10654-023-01020-5>.
- Lu, Mingming, Yadong Zhang, Fengchun Yang, Jialin Mai, Qianwen Gao, Xiaowei Xu, Hongyu Kang, et al. 2023. “TWAS Atlas: A Curated Knowledgebase of Transcriptome-Wide Association Studies.” *Nucleic Acids Research* 51 (D1): D1179–87. <https://doi.org/10.1093/nar/gkac821>.
- Net, Jeroen B. van der, A. Cecile J.W. Janssens, Marinus J.C. Eijkemans, John J.P. Kastelein, Eric J.G. Sijbrands, and Ewout W. Steyerberg. 2008. “Cox Proportional Hazards Models Have More Statistical Power than Logistic Regression Models in Cross-Sectional Genetic Association Studies.” *European Journal of Human Genetics* 16 (9): 1111–16. <https://doi.org/10.1038/ejhg.2008.59>.
- Robinson, Jonathan L., Pinar Kocabaş, Hao Wang, Pierre-Etienne Cholley, Daniel

- Cook, Avlant Nilsson, Mihail Anton, et al. 2020. "An Atlas of Human Metabolism." *Science Signaling* 13 (624): 1–12. <https://doi.org/10.1126/scisignal.aaz1482>.
- Wingo, Thomas S, Yue Liu, Ekaterina S. Gerasimov, Jake Gockley, Benjamin A. Logsdon, Duc M. Duong, Eric B. Dammer, et al. 2021. "Brain Proteome-Wide Association Study Implicates Novel Proteins in Depression Pathogenesis." *Nature Neuroscience* 24 (6): 810–17. <https://doi.org/10.1038/s41593-021-00832-6>.
- Xu, Yu, Scott C. Ritchie, Yujian Liang, Paul R. H. J. Timmers, Maik Pietzner, Loïc Lannelongue, Samuel A. Lambert, et al. 2023. "An Atlas of Genetic Scores to Predict Multi-Omic Traits." *Nature* 616 (7955): 123–31. <https://doi.org/10.1038/s41586-023-05844-9>.

11th Mar 2025

Manuscript Number: MSB-2024-12609R

Title: Metabolic reaction fluxes as amplifiers and buffers of risk alleles for coronary artery disease

Author: Carles Foguet

Xilin Jiang

Scott Ritchie

Elodie Persyn

Yu Xu

Chief Ben-Eghan

Henry Taylor

Emanuele Di Angelantonio

John Danesh

Adam Butterworth

Samuel Lambert

Michael Inouye

Dear Dr Foguet,

Thank you for sending us your revised manuscript. We have now received feedback from two reviewers who were asked to re-evaluate your work. As you will see below, the reviewers are satisfied with the modifications made and support the publication of your manuscript.

Before we can formally accept your manuscript, please address the following editorial level issues:

1. Please provide up to five keywords.
2. Please remove the Author Contribution section from the manuscript.
3. Appendix: title page should contain "Appendix for + manuscript title" and a Table of Content with the page numbers for the listed items.
4. Funding information: all funders should be included in the "More Funders" list rather than the Comments box.
5. "Disclosure statement and competing interests " should be renamed to " DISCLOSURE AND COMPETING INTERESTS STATEMENT".
6. The Appendix datasets S1 and S2 should be renamed to "Dataset EV 1" and "Dataset EV2". The source file names, titles, legends and manuscript callouts all need to be updated accordingly.
7. Section order should be corrected: Title page - Abstract & Keywords - Introduction - Results - Discussion - Methods - Data Availability - Acknowledgements - Disclosure and Competing Interests Statement - References - Figure Legends - Table(s) - Expanded View Figure Legends.
8. Please address the following issues related to figure legends:
 - Please note that the error bars are not defined in the legend of figure 2.
 - Please note that the measure of center for the error bars needs to be defined in the legend of figure 2.

When you resubmit your manuscript, please download our CHECKLIST (<https://bit.ly/EMBOPressAuthorChecklist>) and include the completed form in your submission. *Please note* that the Author Checklist will be published alongside the paper as part of the transparent process (<https://www.embopress.org/page/journal/17444292/authorguide#transparentprocess>)

Click on the link below to submit your revised paper.

Thank you for submitting this interesting paper to Molecular Systems Biology.

Kind regards,
Jingyi

Jingyi Hou, PhD
Senior Editor
Molecular Systems Biology

*** PLEASE NOTE *** As part of the EMBO Press transparent editorial process initiative (see our Editorial at <https://dx.doi.org/10.1038/msb.2010.72> , Molecular Systems Biology will publish online a Review Process File to accompany accepted manuscripts. When preparing your letter of response, please be aware that in the event of acceptance, your cover letter/point-by-point document will be included as part of this File, which will be available to the scientific community. More information about this initiative is available in our Instructions to Authors. If you have any questions about this initiative, please contact the editorial office (msb@embo.org).

Reviewer #1:

The authors have adequately addressed this reviewer's comments. No further comments.

Reviewer #3:

The authors have addressed my comments. I recommend accepting the paper.

All editorial and formatting issues were resolved by the authors.

13th Mar 2025

Manuscript number: MSB-2024-12609RR

Title: Metabolic reaction fluxes as amplifiers and buffers of risk alleles for coronary artery disease

Dear Dr Foguet,

Thank you again for sending us your revised manuscript. I am pleased to inform you that your manuscript has been accepted for publication in Molecular Systems Biology. It has been a pleasure working with you to bring it to this stage.

Yours sincerely,
Jingyi

Jingyi Hou, PhD
Senior Editor
Molecular Systems Biology
